

# A sub-grid model for improving the spatial resolution of air quality modelling at a European scale

Mark R. Theobald[1,2], David Simpson[3,4], Massimo Vieno[5]

[1]Atmospheric Pollution Division, Research Centre for Energy, Environment and Technology (CIEMAT), Madrid, 28040, Spain
[2]Dept. Agricultural Chemistry and Analysis, Higher Technical School of Agricultural Engineering, Technical University of Madrid, 28040, Spain
[3]EMEP MSC-W, Norwegian Meteorological Institute, Oslo, 0313, Norway
[4]Dept. Earth & Space Sciences, Chalmers University of Technology, Gothenburg, SE-412 96, Sweden
[5]Centre for Ecology & Hydrology, Edinburgh Research Station, Penicuik, EH26 0QB, United Kingdom

*Correspondence to*: Mark R. Theobald (mark.theobald@ciemat.es)

**Abstract.** Currently, atmospheric chemistry and transport models (CTMs) used to assess impacts of air quality applied at a European scale lack the spatial resolution necessary to simulate fine-scale spatial variability. This spatial variability is especially important for assessing the impacts to human health or ecosystems of short-lived pollutants, such as nitrogen dioxide ($NO_2$) or ammonia ($NH_3$). In order to simulate this spatial variability, a sub-grid model has been developed to estimate the spatial distributions (at a spatial resolution of $1 \times 1$ km$^2$) of annual mean atmospheric concentrations within the grid squares of a CTM (in this case with a spatial resolution of $50 \times 50$ km$^2$). This is done by combining high spatial resolution emission data with simple parameterisations of atmospheric dispersion. The sub-grid model was tested for two European sub-domains (the Netherlands and central Scotland) and evaluated using $NO_2$ and $NH_3$ concentration data from monitoring networks within each domain. A statistical comparison of the performance of the two models shows that the sub-grid model represents a substantial improvement on the predictions of the CTM, reducing both mean model error (from 60% to 40% for $NO_2$ and from 42% to 26% for $NH_3$ and increasing the spatial correlation (r) with the measured concentrations (from 0.0 to 0.42 for $NO_2$ and from 0.74 to 0.85 for $NH_3$). This improvement was greatest for monitoring locations close to pollutant sources. Although the model ideally requires high spatial resolution emission data, which is not available for the whole of Europe, the use of a Europe-wide emission dataset with a lower spatial resolution also gives an improvement on the CTM predictions for the two test domains. The sub-grid model provides a simple and robust method to estimate sub-grid variability that can potentially be extended to different time scales and pollutants.

**Keywords:** Air quality; Impacts; Atmospheric chemistry and transport models; Sub-grid variability



## 1 Introduction

The impacts of air pollution on human health and natural ecosystems are often evaluated using data from atmospheric dispersion models or atmospheric chemistry and transport models (CTMs). The scale of these evaluations ranges from local assessments with domains of several kilometres (e.g. Dragosits et al., 2002, Aggarwal and Jain, 2015, Galvis et al., 2015) to global assessments using grid cells of 1–10 degrees (see for example Dentener et al., 2006). The spatial resolution used in these assessments depends on many factors, including availability of input data, model assumptions, receptor type (e.g. people, forests, etc) and computation time. Many of the impact assessments at a European scale are carried out using atmospheric concentration or deposition predictions of the model developed by the Meteorological Synthesizing Centre-West (MSC-W) of the European Monitoring and Evaluation Programme (EMEP). The EMEP MSC-W model (Simpson et al., 2012), called the EMEP model hereafter, has commonly been applied for policy purposes at a spatial resolution of ca. 50 $\times$ 50 km$^2$ (e,g. Fagerli and Aas, 2008, Simpson et al., 2006). Although the model is increasingly used at even finer resolution (e.g. 0.1 $\times$ 0.1 degrees) even for official MSC-W purposes (EMEP, 2015), such runs are extremely CPU intensive for European-scale modelling, and cannot be used for the 100s–1000s of simulations required by the source-receptor matrices which are an important output of MSC-W (EMEP, 2015). EMEP model results also underpin the Greenhouse gas - Air pollution Interactions and Synergies (GAINS) model, which is a key tool in developing European policy within both UN-ECE and the European Union (Amann et al., 2011). However, the resolution of the EMEP model (or any other European scale CTM), at least when run in typical policy mode, is not currently high enough to resolve the large horizontal concentration gradients found close to sources of relatively short-lived pollutants, such as ammonia ($NH_3$), nitrogen dioxide ($NO_2$) or sulphur dioxide ($SO_2$) (CLRTAP, 2014; Denby et al., 2011).

The EMEP model predicts the mean atmospheric concentration within each grid square. However, within a grid square there may be concentrations an order of magnitude (or more) above and below this mean value, even if the mean prediction is correct. Neglecting this sub-grid variability (SGV) can strongly bias assessments of air pollution impacts. For example, Denby et al. (2011) estimated that urban background exposure to $NO_2$ is underestimated by an average of 44% when the 50 $\times$ 50 km$^2$ grid concentrations of the EMEP model are used. This problem is not restricted to the low grid resolution used by the EMEP model, it also occurs in assessments with higher resolutions. For example, Hallsworth et al. (2010) used a CTM to estimate $NH_3$ concentrations in the UK at spatial resolutions of 5 $\times$ 5 km$^2$ and 1 $\times$ 1 km$^2$. They found that the 5 km model estimated that the $NH_3$ critical level of 1 ug m$^{-3}$ was exceeded for 40% of the total area of UK Special Areas of Conservation (SAC), whereas the 1 km model estimated an exceedance for only 21%. This reduction in the area of exceedance when the model resolution was increased was due to the ammonia sources (agricultural areas) and the SAC being separated spatially. Modelling at a higher resolution resolved the large horizontal concentration gradients better, thus predicting higher concentrations in the agricultural areas and lower concentrations within the SAC. By contrast, Oxley and ApSimon (2007) found that increasing the model spatial resolution from 50 km to 5 km and from 5 km to 1 km increased the estimates of exposure to primary particles with a diameter of 10 µm or less ($PM_{10}$) in urban areas. This is because in this case the urban



areas are also some of the largest sources of primary $PM_{10}$. A multi-model study involving five CTMs to simulate pollutant concentrations across Europe found a large increase in annual mean concentration predictions of $PM_{10}$ and $NO_2$ in urban locations when increasing the spatial resolution through the range 56, 28, 14 and 7 km (Cuvelier et al., 2013; Schaap et al., 2015). For most of the models, about 70% of the model response to the change of resolution was due to the change in the spatial distribution of emissions. By comparing the concentration predictions in urban areas with measured values, model performance (slope, bias and correlation) was generally found to improve for all models as the resolution was increased. In order to resolve the large horizontal concentration gradients found in urban areas, Cuvelier et al. (2013) suggested that a resolution of a few km down to one km would be needed, but added that this is not currently feasible for application across Europe. However, even this might not be sufficient for resolving the large horizontal concentration gradients of $NO_2$, for example.

Several potential methods could be used to estimate the SGV of the concentration predictions of short-lived air pollutants across Europe. Firstly, the EMEP model could be applied at a higher resolution. This has been done in the UK for a resolution of $5 \times 5$ km$^2$ (EMEP4UK) (Vieno et al., 2010; Vieno et al., 2014), and for Europe at ca. $7 \times 7$ km$^2$ (Schaap et al., 2015, EMEP, 2015), but such runs are extremely CPU demanding. A European application at $1 \times 1$ km$^2$ resolution or higher is currently not feasible. Such runs would also require a consistent and accurate high resolution emission dataset, which is not currently available. A second solution is the 'stitching together' of national modelling simulations at a high resolution (see, for example, de Smet et al., (2013); Janssen and Thunis, 2016). This approach has the advantage of making use of national expertise and emission and meteorological datasets. However, the disadvantages are that it is likely to lead to 'border effects' as a result of differing methodologies and/or input datasets used by neighbouring countries and results may not be available for all countries, making it difficult to carry out a consistent assessment for the whole of Europe. The third solution is to apply geo-statistical techniques to the low resolution concentration data (e.g. from the EMEP model) that makes use of other relevant spatial datasets. These techniques can be used to either estimate the probability distribution of the concentration (or a related quantity) within each grid square or to explicitly estimate the spatial distribution of the concentration within the grid square. An example of the former approach is that of Denby et al. (2011) who estimated the population-weighted concentrations of $NO_2$, $PM_{10}$ and $O_3$ within each EMEP $50 \times 50$ km$^2$ grid square using information on measured concentrations and their covariance with population density, that was then parameterised using emission and altitude data. Another example is the SGV parameterisation of Ching et al. (2006) for the CMAQ model based on sub-grid concentration distributions of benzene and formaldehyde calculated using the ISCST3 short-range dispersion model. The same CMAQ simulations were used by Isakov et al. (2007) to develop a method to explicitly model the sub-grid spatial distributions of concentrations at a resolution of $200 \times 200$ m$^2$. Their method used relationships between the sub-grid concentrations and sub-grid emission strengths derived from short-range dispersion modelling results, although it was only applied to a small area (Philadelphia County). A different geo-statistical approach was used by Janssen et al. (2012), in which they estimated sub-grid concentrations for Belgium by using empirical relationships between long-term atmospheric concentrations and land use characteristics. A Europe-wide approach was developed for $NO_2$ and particulate matter by



Kiesewetter et al. (2013 and 2014), although only at a resolution of $7 \times 7$ km$^2$. In their work, concentrations simulated by the EMEP model at a resolution of $28 \times 28$ km$^2$ were disaggregated using an 'urban increment'. This increment was calculated from the concentration predictions of the CHIMERE model (Bessagnet et al., 2004) at a resolution of $7 \times 7$ km$^2$. The relationship between the differences in the concentration predictions of the two models and the emission rate (from near-ground-level sources only) used for each 7 km grid square was used to calculate the urban increment. Model evaluation using annual mean concentrations from more than 1500 urban background monitoring stations showed that the model can predict concentrations within a factor of two of the measured value for most locations. The authors also developed a parameterisation to estimate the additional concentration increment at the locations of roadside air quality stations, although this approach relies heavily on measurement data.

In this paper we present the development, testing and evaluation of a simple sub-grid model that combines high-spatial-resolution emission data and a simple parameterisation of short-range dispersion to estimate the spatial distribution of concentrations of short-lived pollutants within the EMEP model grid squares. The sub-grid model is used to calculate the annual mean concentrations of NO$_2$ and NH$_3$ for 2008 at a resolution of $1 \times 1$ km$^2$ for two test domains (central Scotland and the Netherlands) and evaluated using monitoring network data from within the two domains. Section 2 provides information on the methods and datasets used and Section 3 describes the model development process. Section 4 presents the results of the sub-grid modelling, model evaluation and an analysis of the sensitivity of the model to some of the parameters and datasets used whilst Section 5 discusses model performance, its applicability, and potential improvements and extensions.

## 2 Materials and Methods

The two domains used in this study are central Scotland and the Netherlands (Fig. 1). These domains were chosen because they provide a contrast between a built-up, industrialised and agricultural region (the Netherlands) and a region with both large cities, intensive industrial and agricultural areas as well as more extensively used or semi-natural areas (central Scotland). Both domains also have NH$_3$ and NO$_x$ emission inventory data at a ca. $1 \times 1$ km$^2$ resolution. Spatially distributed annual NH$_3$ and NO$_x$ emission data for the study year (2008) were obtained from the National Atmospheric Emissions Inventory (http://naei.defra.gov.uk/) for the Scottish domain and from the National Institute for Public Health and the Environment (RIVM), for the Netherlands (Fig. 1). In order to evaluate the sub-grid model for an emission dataset with a lower spatial resolution that could be used for a Europe-wide application of the sub-grid model, the 2008 'EC4MACS' emissions with a spatial resolution of ca. $7 \times 7$ km$^2$ (EC4MACS, 2012, also used in Schaap et al., 2015) were also used for the two domains.

Meteorological data used to develop the sub-grid model were taken from the Lyneham meteorological station in the UK for 1995 (LYNE95) (Spanton et al., 2004), which was a fairly typical year with regards to mean air temperature and wind speed. Although an arbitrary choice, this dataset was chosen because it has been used in various model evaluation studies and has been made freely available to the dispersion modelling community (e.g. Hall et al., 2000; Theobald et al., 2012). In order to



make the dataset less location-specific (LYNE95mod) the wind direction data were randomised and the wind speed was scaled so that the annual mean value was equal to the annual domain mean value used in the EMEP model for the 2008 study year (5.1 m s$^{-1}$). The use of a single UK meteorological dataset from a different year to the study year for the development of a model applied at the European scale may introduce a large amount of error or uncertainty in the predictions. In order to
assess this uncertainty, two domain-specific meteorological datasets for the study year were also tested. These datasets were from Easter Bush, for Scotland (von Bobrutzki et al., 2010), and Cabauw, for the Netherlands (obtained from the Cesar Database: http://www.cesar-database.nl/).

In order to parametrise the pollutant dispersion from source areas, three different atmospheric dispersion models were used.
These were ADMS (v4) (Carruthers et al., 1994), AERMOD (v12345) (Cimorelli et al., 2002) and LADD (Dragosits et al., 2002). These three models were chosen because they have been extensively evaluated for the atmospheric dispersion of NO$_2$ and NH$_3$, with the exception of LADD, which has only been evaluated for NH$_3$ (Theobald et al., 2012).
Model evaluation was carried out using 2008 annual mean concentration data from local and national monitoring networks in the two study domains. For Scotland, NO$_2$, data were obtained from the Air Quality in Scotland website
(http://www.scottishairquality.co.uk/) (48 stations: 37 traffic and 11 non-traffic sites) and from RIVM for the Netherlands (43 stations: 13 traffic and 30 non-traffic). The evaluation was done for all sites, and for the traffic and non-traffic sites separately since the traffic sites are strongly influenced by the exact site location and are unlikely to be representative of a 1 × 1 km$^2$ grid square. For NH$_3$ concentrations in the Scottish domain, monitoring data were obtained from the UK National Ammonia Monitoring Network (NAMN) (Sutton et al., 2001) (http://uk-air.defra.gov.uk/networks/network-info?view=nh3),
which has 14 sites within the domain. In addition, NH$_3$ monitoring data from 21 sites in a local network covering 36 km$^2$ (Vogt et al., 2013) were also used. For the Netherlands, NH$_3$ concentration data from the Measuring Ammonia in Nature (MAN) network (Lolkema et al., 2015) were provided by RIVM (108 stations). Model performance was assessed using the evaluation statistics of the R package "OpenAir" (Carslaw and Ropkins, 2012). Four performance metrics were used to compare the modelled concentrations with the observed values: fraction of model predictions within a factor of two of the
observations (FAC2), normalised mean bias (NMB), normalised mean gross error (NMGE) and the Pearson correlation coefficient (r) (see Appendix A for definitions).

## 3 Model development

The sub-grid 1 × 1 km$^2$ concentration estimates were calculated from three components: the EMEP 50 × 50 km$^2$ concentration predictions, the 1 × 1 km$^2$ emission data and an estimate of short-range (< 50 km) pollutant dispersion. Figure
2 shows a schematic of the process. Short-range pollutant dispersion was parameterised using a simple scenario of a single 1 × 1 km$^2$ source with an emission rate of 1 Mg km$^{-2}$ yr$^{-1}$ in the centre of a square domain (of dimensions 101 × 101 km$^2$). For the dispersion of NH$_3$, the source was assumed to be at ground level (a suitable approximation for most agricultural sources,





which account for more than 90% of emissions in Europe). For $NO_2$, the assumption was made that $NO_2$ concentrations were linearly correlated with $NO_x$ concentrations (valid for annual mean concentrations). This allowed us to use the $NO_x$ emissions for the calculation of $NO_2$ concentrations without considering photochemical reactions. Emissions of $NO_x$ can occur over a range of emission heights, depending on the source type. Since the emission height will affect the resulting

$NO_2$ concentrations at ground level, it needs to be taken into account. This was done by assigning a representative emission height for each emission sector (Selected Nomenclature for Air Pollution (SNAP) code) that contributed more than 1% of the total domain emissions (Table 1). These emission heights correspond loosely to the mean effective emission heights used in the EMEP model for the sector emissions. In order to test the sensitivity of the sub-grid model to the emission heights used, additional simulations were carried out using emission heights half and double these values. For the ground level source, all

three dispersion models (ADMS, AERMOD and LADD) were used to simulate the annual mean near-ground-level concentrations of $NH_3$ and $NO_2$ on a 1 km grid (for the $101 \times 101$ km$^2$ domain) using the LYNE95mod meteorological dataset. For the elevated source scenarios, only ADMS and AERMOD were used to simulate the annual mean concentrations because the LADD model is not suitable for simulating dispersion from elevated sources (Theobald et al., 2012). A height of 1.5 m was used for the near-ground-level concentrations, because this height is commonly used for

concentration monitoring and impact assessments (Cape et al., 2009). No removal processes (chemical reactions, dry or wet deposition etc.) were simulated because these processes depend strongly on local conditions (concentrations of other chemical species, meteorological conditions, surface characteristics, etc.). The result of these simulations was nine concentration fields, three for ground level sources (three models $\times$ one source height) and six for elevated sources (two models $\times$ three source heights) centred on the source location. For each source height, a rotationally symmetric concentration

field (or kernel) was obtained by fitting regression curves to the modelled concentrations (natural log of concentrations vs. natural log of distance from source centre), which was then averaged over all models (more details are provided in the supplementary material).

These kernels were then multiplied by the emission data (for each SNAP sector separately in the case of $NO_x$) using a "moving window" approach and the results summed over the entire domain (central Scotland or the Netherlands). The

resulting "sub-grid distributions" provide an estimate of the spatial variability of the concentrations at a $1 \times 1$ km$^2$ resolution, which were then used to "redistribute" the EMEP predictions within each $50 \times 50$ km$^2$ grid square. This step is necessary since the sub-grid model does not take into account large scale processes such as long-range transport or chemical transformations of pollutants, processes that are included in the large scale model (the EMEP model, in this case). The simplest way to do this redistribution would be to multiply the sub-grid distributions by the EMEP predictions and then

divide by the mean value of the sub-grid distribution for each $50 \times 50$ km$^2$ grid square. This approach conserves the sub-grid distribution for each $50 \times 50$ km$^2$ square and also has the same mean concentration as the EMEP prediction. However, it also could lead to large discontinuities at the edges of the EMEP grid squares if the ratio between the mean of the sub-grid distribution and the EMEP prediction differ greatly from that of adjacent squares. To avoid this problem, the ratio of the EMEP predictions to the mean value of the sub-grid distribution for each $50 \times 50$ km$^2$ square was interpolated on a $1 \times 1$ km$^2$



grid (using a spline interpolation of the values at the centre of each grid square in ArcGIS 10.2 (Environmental Systems Research Institute, Redlands, CA, USA)). The interpolated field was then multiplied by the sub-grid distribution and then the process was repeated over ten iterations. In fact only four-five iterations were necessary to give concentration fields that differed by a maximum of 1%. A more detailed description of the process is provided in the supplementary material.

## 4 Results

### 4.1 Sub-grid concentration predictions and model evaluation

Figure 3 shows the sub-grid concentration predictions for $NO_2$ and $NH_3$ for the two domains. The EMEP concentration fields are also shown for comparison. Table 2 shows the evaluation statistics of the EMEP and sub-grid models for annual mean $NO_2$ concentrations for the Dutch and Scottish monitoring data. In general the sub-grid model performed notably better than the EMEP model as a result of a consistent underestimation by the latter (negative NMB). The mean error of the EMEP model is largest for the Scottish dataset with a NMGE of 82% and 70% for the datasets with and without traffic stations, respectively. The model performs worst for the Scottish traffic stations with a mean underestimation of 84%. The EMEP model performs considerably better for the Dutch dataset, with 91% of predictions within a factor of two of the observed values, although this drops to 69% when considering the traffic stations only. The sub-grid model (using 1 x 1 $km^2$ emissions) also performed best for the Dutch dataset, with a smaller mean bias and error and a better correlation than the EMEP model. However, the EMEP model had a lower mean bias and error for the non-traffic stations. The sub-grid model also performed better than the EMEP model for the Scottish dataset (both with and without traffic stations), as well as for the combined dataset (Netherlands plus Scotland). Similarly to the EMEP model, the sub-grid model performed worst for the Scottish traffic stations, although notably better than the EMEP model. The use of the lower resolution emissions actually improved the performance of the sub-grid model for some of the statistics (most notably for the non-traffic stations in the Netherlands domain).

Table 3 shows the evaluation statistics of the EMEP and sub-grid models for annual mean $NH_3$ concentrations for the Dutch and Scottish monitoring data. In general the sub-grid model performed notably better than the EMEP model. The EMEP model performed worse for the local monitoring network, as all monitoring locations were within a single EMEP $50 \times 50$ $km^2$ square. The sub-grid model (using 1 x 1 $km^2$ emissions) also performed worst for this dataset, although its performance was better than that of the EMEP model, as it was for all the datasets except for the National Ammonia Monitoring Network sites. The use of the 7 x 7 $km^2$ emissions worsened the performance of the sub-grid model for all datasets except for the National Ammonia Monitoring Network sites, for which it had a similar performance to the model using the higher resolution emissions. Figure 4 shows the scatterplots of $NO_2$ and $NH_3$ concentration predictions of the EMEP and sub-grid model vs. the observed values for all sites in both domains.



### 4.2 Sensitivity of sub-grid model predictions to model parameters

The use of domain-specific meteorological datasets only had a small effect on the concentration estimates of the sub-grid model (Fig. 5). Mean differences from the estimates using the generic meteorological dataset (LYNE95mod) were 7% for both $NO_2$ and $NH_3$, although differences of up to 29% were found for individual measurement sites. Model performance was

barely affected (not shown). Randomising the wind direction data of the domain-specific datasets gave very similar results to those using the generic dataset, with maximum differences of only 1% (not shown). This suggests that the meteorological factor that most influences the sub-grid model estimates is the wind direction distribution.

The sub-grid model estimates are also not very sensitive to the $NO_x$ emission height. On average, the effect on the concentration predictions of halving or doubling the emission heights is less than 2%, with a maximum difference of 6%.

This lack of sensitivity to the exact height used reflects the fact that ground-level sources contribute significantly more to near-source concentrations than elevated sources.

### 5 Discussion

### 5.1 An improvement, but is it enough?

These results show that a simple and robust geostatistical approach can be used to improve the EMEP model predictions of

$NO_2$ and $NH_3$ annual concentrations. This improvement is not surprising considering the large difference in spatial resolutions (50 km vs. 1 km) and the strong link between short-lived pollutants and the spatial distribution of emissions. However, is this improvement large enough to warrant the inclusion of such a sub-grid model into the output processing options of a chemical transport model? In order to answer this question, we can use the concept of model acceptability suggested by Chang and Hanna (2004). This concept can be used to evaluate whether the EMEP model and/or the sub-grid

model perform acceptably and, therefore, whether the sub-grid model represents an improvement on the EMEP model alone, in terms of model acceptability. Hanna and Chang (2012) suggested that an 'acceptable' model is one that meets the criteria for more than half of a series of statistical tests. The performance metrics used are: fractional bias, geometric mean bias, normalised mean square error, geometric variance and FAC2 (see Appendix A for definitions and acceptability criteria). In the current study, we define an acceptable model as one that meets at least three of these five criteria (for each dataset).

Although the concept of model acceptability of Chang and Hanna (2004) was defined for research-grade experimental data, the fact that we are considering annual mean concentrations (instead of high temporal resolution measurements), should make the approach suitable for use with operational models and monitoring data, such as those used here. For the two combined datasets ($NO_2$-All and $NH_3$-All) shown in Fig. 4, the EMEP model meets none and five of the five criteria for $NO_2$ and $NH_3$ respectively, whereas the sub-grid model meets three and five criteria, respectively (Table 4). This suggests that

the sub-grid model is a significant improvement (in terms of model acceptability) for $NO_2$ (even when the lower resolution emission dataset is used), but not for $NH_3$. This can be explained by looking at the number of criteria met for the individual



datasets (Table 4). For $NO_2$, The EMEP model performed acceptably for the Netherlands (All) but not for Scotland (All). This is partly due to the Dutch network having a larger proportion of non-traffic sites (70% vs. 23%), which would be more representative of the $50 \times 50$ km$^2$ grid cells. However, the EMEP model also performed acceptably for the Dutch traffic stations but neither the EMEP model nor the sub-grid model performed acceptably for the Scottish traffic stations. Looking

more carefully at the traffic stations used in the domains reveals that station siting may have an influence on model performance. According to the information available regarding the Scottish traffic sites, monitoring stations are located between 0.5 and 16 m from the road edge. Although no information is available regarding the exact locations of the Dutch monitoring stations, Nguyen et al. (2012) point out that one station in the Amsterdam Municipal Health Service (GGD) network (not used in this study) "is very close to the road (< 2.5 m)". This suggests that, in general, sites in the Dutch

network are > 2.5 m from the road, whereas in the Scottish network 17 of the 37 traffic sites are closer than this. This difference in station siting could be the reason why neither the EMEP nor sub-grid model performs acceptably for the Scottish dataset. For $NH_3$, the EMEP and sub-grid model perform acceptably for the two national networks but only the sub-grid model performs acceptably for the local network. This is probably because the national networks site their monitoring stations far from the influence of individual emission sources in order to be representative of a large area, whereas the local

network was located in an area with intensive poultry farming and was designed to assess the influence of individual sources. Since the majority (86%) of the sites used in the analysis belonged to the national networks, overall model performance was similar to model performance for those networks. The sub-grid approach, therefore, is most useful where there are large horizontal concentration gradients, such as within large cities (for $NO_2$) or areas with intensive agriculture (for $NH_3$), which is where the largest impacts are most likely to occur.

It is also worth briefly comparing the improvements in model performance with those reported by other studies. Denby et al. (2011) showed that the population weighted concentration for $NO_2$ was, on average, 44% higher with their sub-grid parameterisation than that calculated using the original concentrations from the EMEP model. Although not directly comparable (since we do not calculate population weighted concentrations), $NO_2$ concentrations estimated using our sub-grid model were, on average 77% higher than those of the EMEP model at the monitoring station locations. Despite this increase,

the sub-grid model estimates were still, on average, 27% lower than the measured concentrations. Janssen et al. (2012) showed that their approach of downscaling modelled concentrations from $15 \times 15$ km$^2$ to $3 \times 3$ km$^2$ reduced model error by about 20%. Our sub-grid model for $NO_2$ reduced model error by 30–40%, although for a larger change in resolution ($50 \times 50$ km$^2$ to $1 \times 1$ km$^2$). In the study by Schaap et al. (2015), increasing the spatial resolution from approx. $56 \times 56$ km$^2$ to $7 \times 7$ km$^2$ increased the correlation (r) between the models' predictions and hourly urban background $NO_2$ concentrations from

approx. 0.1–0.4 to 0.6–0.7 and reduced model bias by approx. 60–90% for most of the models. For a similar change in spatial resolution ($50 \times 50$ km$^2$ to $7 \times 7$ km$^2$), our sub-grid model for annual mean $NO_2$ concentrations using the low resolution emissions increased r from 0.16–0.54 to 0.51–0.79 and reduced model bias by approx. 20–70%.



## 5.2 How can the sub-grid approach be applied?

Two potential uses of the sub-grid approach can be envisaged: a Europe-wide application to provide a spatial assessment of exceedance of $NO_2$ and $NH_3$ annual limit values or critical levels and the assessment of individual emission hot-spots in areas where detailed modelling assessments are not available but high resolution emission data are. In the latter case, if the hot-spot domain is located within a single EMEP $50 \times 50$ km$^2$ grid square, the smoothing step would not be necessary. The Europe-wide application would require high spatial resolution emission data for the whole domain. There is, as far as we are aware, currently no European emission inventory with a spatial resolution close to $1 \times 1$ km$^2$. The highest resolutions available are the $7 \times 7$ km$^2$ emission inventories produced for various EU projects (Kuenen et al., 2014, EC4MACS, 2012). As shown above, the use of emission data at this resolution still gives an improvement on the concentration predictions and even performs better than the sub-grid model using the higher resolution emissions, in some cases.

## 5.3 Advantage, disadvantages and potential improvements

The sub-grid model can provide more accurate concentration predictions than the EMEP model alone, especially close to emission sources. However, this approach has only been tested for annual mean $NO_2$ and $NH_3$ concentrations, although could potentially be extended to other short-lived pollutants and shorter time scales (daily or hourly). This means that the model cannot currently be used to assess exceedance of short-term limit values (e.g. for Europe, an hourly mean concentration of 200 µg $NO_2$ m$^{-3}$ more than 18 times in one year) although, as shown by Kiesewetter et al. (2013), the annual mean limit values for $NO_2$ and $PM_{10}$ are more stringent targets. Critical levels for ammonia are expressed as annual mean concentrations and so a sub-grid model with a higher temporal resolution is not necessary. The other limitation of the approach is the need for high resolution emission data although, as shown above, the use of emission data with a resolution of $7 \times 7$ km$^2$ already produces improvements in model performance compared with the original CTM concentration estimates.

With regards to potential improvements, in addition to the extension to shorter time periods, it would also be possible to include spatially-varying wind data since this has the potential to better represent local conditions. Such data could be obtained directly from the meteorological fields of the CTM. It also would be possible to incorporate stack parameters (effective emission heights and the contribution of stack emissions to the emissions of a particular grid square) from officially reported data and/or other data sources. This would potentially improve concentration estimates close to large stack sources. As shown above, model performance is poorer for sites very close to roads and so the inclusion of a roadside increment model could also improve the model estimates. However, by increasing the complexity of the model, we have to careful not to lose sight of the objective of the sub-grid model, which is to provide a robust and simple method of post-processing concentrations estimated by a chemical transport model.

The sub-grid approach also has the potential to be applied to other pollutants for which there is a strong relationship between emissions and concentrations. Zhang and Wu (2013) analysed air quality simulations of the CMAQ model to quantify the



influence of a range of processes on the atmospheric concentrations of several pollutants. The species that were most strongly influenced by emission processes were: $NH_3$, $NO$, $NO_2$, $SO_2$, $PM_{2.5}$, $SO_2^{4-}$, elemental carbon, and primary organic aerosol and are, therefore, potential candidates for an extension of the model. The spatial distribution of ozone, a secondary pollutant, cannot be estimated based on emissions but its inverse relationship with $NO_x$ could be exploited to model the sub-

grid variability. Apart from concentrations, it may also be possible to develop a sub-grid model for processes such as wet deposition of nitrogen or sulphur, for which high resolution rainfall maps could be used to estimate the sub-grid distributions. Dry deposition of reduced nitrogen could also be modelled using the $NH_3$ concentration distribution and land cover parameters, assuming that most of the deposition is in the form of $NH_3$. Dry deposition of oxidised nitrogen would be more difficult since there is no one dominant species that contributes.

**Conclusions**

The sub-grid spatial variability of the annual mean concentration predictions of $NO_2$ and $NH_3$ of an atmospheric chemistry and transport model can be estimated by combining the predictions with high spatial resolution emission datasets and short-range dispersion fields. This paper describes the development of this technique and its application to two test domains in Europe. Comparison of the sub-grid model predictions with annual mean concentrations measured within both domains

shows that the sub-grid model represents an improvement on the predictions of the chemical transport model reducing both model error and bias and increasing the spatial correlation with the measured concentrations.

**Code availability**

The sub-grid model code (in R) plus example input and output files are provided in the supplementary material.

**Data availability**

The data shown in Fig. 4 and Fig. 5 are provided in the supplementary material.





## Appendix A: Descriptions of the performance metrics used

**Table A1: The four metrics relating modelled concentrations ($M_i$) with the observed values ($O_i$), used for evaluating model performance.**

| Performance measure | Definition |
|---|---|
| Fraction of model predictions within a factor of two of the observations (FAC2): | $0.5 \leq \dfrac{M_i}{O_i} \leq 2.0$ |
| Normalised mean bias: | $NMB = \dfrac{\sum\limits_{i=1}^{n} M_i - O_i}{\sum\limits_{i=1}^{n} O_i}$ |
| Normalised mean gross error: | $NMGE = \dfrac{\sum\limits_{i=1}^{n} \left| M_i - O_i \right|}{\sum\limits_{i=1}^{n} O_i}$ |
| Pearson correlation coefficient: | $r = \dfrac{1}{(n-1)} \sum\limits_{i=1}^{n} \left( \dfrac{M_i - \overline{M}}{\sigma_M} \right) \left( \dfrac{O_i - \overline{O}}{\sigma_O} \right)$ |

5 **Table A2: The five performance measures relating modelled concentrations ($M_i$) with the observed values ($O_i$) used to assess model acceptability.**

| Performance measure | Definition | Optimum value | Acceptability Criterion |
|---|---|---|---|
| Fractional bias (FB) | $FB = \dfrac{2\left( \overline{O} - \overline{M} \right)}{\left( \overline{O} + \overline{M} \right)}$ | 0 | $|FB| < 0.3$ |
| Geometric Mean Bias (MG) | $MG = \exp\left( \overline{\ln O} - \overline{\ln M} \right)$ | 1 | $0.7 < MG < 1.3$ |
| Normalised mean square error (NMSE) | $NMSE = \dfrac{\overline{\left( O - M \right)^2}}{\overline{O}\,\overline{M}}$ | 0 | $NMSE < 1.5$ |
| Geometric variance (VG) | $VG = \exp\left[ \overline{\left( \ln O - \ln M \right)^2} \right]$ | 1 | $VG < 4$ |
| Fraction of model predictions within a factor of two of the observations (FAC2) | $0.5 \leq \dfrac{M_i}{O_i} \leq 2.0$ | 1 | $FAC2 > 0.5$ |





**Author contribution**

Model development and evaluation was principally carried out by M. Theobald with contributions on the design of the sub-grid modelling methodology from D. Simpson and M. Vieno. Emission datasets were prepared by M. Vieno and the manuscript was prepared by M. Theobald with contributions from both co-authors.

**Acknowledgements**

The work presented in this paper was funded through the EU projects: NitroEurope IP (Project No.: 017841) and ECLAIRE (Project No.: 282910), and EMEP under UN-ECE. We would like to thank Sim Tang at CEH Edinburgh for providing the NAMN data from the Defra project AQ0647 "UK Eutrophying and Acidifying Atmospheric Pollutants (UKEAP), Mhairi Coyle at CEH Edinburgh for providing the Easter Bush meteorological data, the Cabauw Experimental Site for Atmospheric
Research (Cesar) database for the Cabauw meteorological data, Roy Wichink Kruit at RIVM for the Dutch $NO_2$ concentration data and Dorien Lolkema at RIVM for the Dutch $NH_3$ concentration data.

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

**Table 1: Emission heights used for each main emission sector**

| SNAP Code | Emission sector | Effective emission height used (m) |
|---|---|---|
| 1 | Combustion in energy and transformation industries | 400 |
| 2 | Non-industrial combustion plants | 0 |
| 3 | Combustion in manufacturing industry | 400 |
| 4 | Production processes | 50 |
| 7 | Road transport | 0 |
| 8 | Other mobile sources and machinery | 0 |
| 9 | Waste treatment and disposal | 200 |

**Table 2: Performance evaluation of the EMEP and sub-grid models for annual mean $NO_2$ concentrations. The best performing model for each statistic is highlighted in bold. FAC2 is the fraction of model predictions within a factor of two of the observations, NMB is the normalised mean bias, NMGE is the normalised mean gross error and r is the Pearson correlation coefficient. Shaded cells highlight the model performance for the sub-grid model using the lower resolution emission data.**

| Dataset | n | EMEP model | | | | Sub-grid model | | | | |
|---|---|---|---|---|---|---|---|---|---|---|
| | | FAC2 | NMB | NMGE | r | Emission data | FAC2 | NMB | NMGE | r |
| Netherlands (All) | 43 | 0.91 | -0.24 | 0.31 | 0.54 | $1 \times 1$ km$^2$ | **1.0** | **0.04** | 0.22 | **0.84** |
| | | | | | | $7 \times 7$ km$^2$ | **1.0** | -0.08 | **0.21** | 0.79 |
| Netherlands (No traffic stations) | 30 | **1.00** | -0.06 | **0.18** | 0.73 | $1 \times 1$ km$^2$ | **1.0** | 0.07 | 0.27 | **0.86** |
| | | | | | | $7 \times 7$ km$^2$ | **1.0** | **0.01** | 0.21 | 0.81 |
| Netherlands (Traffic stations only) | 13 | 0.69 | -0.45 | 0.45 | 0.17 | $1 \times 1$ km$^2$ | **1.0** | **0.00** | **0.17** | **0.58** |
| | | | | | | $7 \times 7$ km$^2$ | **1.0** | -0.18 | 0.21 | 0.32 |
| Scotland (All) | 48 | 0.06 | -0.82 | 0.82 | 0.16 | $1 \times 1$ km$^2$ | **0.48** | **-0.48** | **0.52** | 0.46 |
| | | | | | | $7 \times 7$ km$^2$ | 0.23 | -0.63 | 0.63 | **0.51** |
| Scotland (No traffic stations) | 11 | 0.27 | -0.70 | 0.70 | 0.40 | $1 \times 1$ km$^2$ | **0.91** | **-0.07** | **0.30** | 0.80 |
| | | | | | | $7 \times 7$ km$^2$ | 0.64 | -0.38 | 0.39 | **0.85** |
| Scotland (Traffic stations only) | 37 | 0.00 | -0.84 | 0.84 | 0.05 | $1 \times 1$ km$^2$ | **0.35** | **-0.54** | **0.55** | 0.50 |
| | | | | | | $7 \times 7$ km$^2$ | 0.11 | -0.67 | 0.67 | **0.51** |
| All | 91 | 0.46 | -0.58 | 0.60 | -- | $1 \times 1$ km$^2$ | **0.73** | **-0.27** | **0.40** | **0.42** |
| | | | | | | $7 \times 7$ km$^2$ | 0.59 | -0.40 | 0.46 | 0.27 |



**Table 3: Performance evaluation of the EMEP and sub-grid models for annual mean NH$_3$ concentrations. The best performing model for each statistic is highlighted in bold. FAC2 is the fraction of model predictions within a factor of two of the observations, NMB is the normalised mean bias, NMGE is the normalised mean gross error and r is the Pearson correlation coefficient. Shaded cells highlight the model performance for the sub-grid model using the lower resolution emission data.**

| Dataset | n | EMEP | | | | Sub-grid model | | | | |
|---|---|---|---|---|---|---|---|---|---|---|
| | | FAC2 | NMB | NMGE | r | Emission data | FAC2 | NMB | NMGE | r |
| Netherlands | 108 | 0.85 | 0.23 | 0.39 | 0.69 | $1 \times 1$ km$^2$ | **0.93** | **0.10** | **0.22** | **0.85** |
| | | | | | | $7 \times 7$ km$^2$ | 0.90 | 0.28 | 0.40 | 0.71 |
| Scotland – Local network | 21 | 0.52 | -0.47 | 0.65 | -- | $1 \times 1$ km$^2$ | **0.62** | **-0.08** | **0.54** | **0.48** |
| | | | | | | $7 \times 7$ km$^2$ | 0.52 | -0.48 | 0.66 | -- |
| Scotland (National Ammonia Monitoring Network) | 14 | **0.71** | **0.07** | 0.46 | 0.73 | $1 \times 1$ km$^2$ | 0.50 | 0.24 | **0.42** | 0.80 |
| | | | | | | $7 \times 7$ km$^2$ | 0.57 | **0.07** | 0.43 | **0.81** |
| All | 143 | 0.79 | 0.17 | 0.42 | 0.74 | $1 \times 1$ km$^2$ | **0.84** | **0.09** | **0.26** | **0.85** |
| | | | | | | $7 \times 7$ km$^2$ | 0.81 | 0.20 | 0.42 | 0.75 |

5 **Table 4: Number of model acceptability criteria met for each model and evaluation dataset. Shaded cells represent acceptable model performance ( ≥ 3 criteria met).**

| Pollutant | Dataset | | No. of criteria met | | |
|---|---|---|---|---|---|
| | | | EMEP | Sub-Grid ($1 \times 1$ km$^2$ emissions) | Sub-Grid ($7 \times 7$ km$^2$ emissions) |
| NO$_2$ | Netherlands | All | 5 | 5 | 5 |
| | Netherlands | No traffic stations | 5 | 5 | 5 |
| | Netherlands | Traffic stations only | 3 | 5 | 5 |
| | Scotland | All | 0 | 2 | 1 |
| | Scotland | No traffic stations | 0 | 5 | 3 |
| | Scotland | Traffic stations only | 0 | 2 | 0 |
| | All | | 0 | 3 | 3 |
| NH$_3$ | Netherlands | | 5 | 5 | 5 |
| | Scotland | Local network | 2 | 5 | 2 |
| | Scotland | National Network | 5 | 5 | 5 |
| | All | | 5 | 5 | 5 |





**Figure 1: Spatial distributions of annual emissions of NO$_x$ (left) and NH$_3$ (right), for the Dutch (top) and Scottish (bottom) domains. The EMEP 50 × 50 km$^2$ grid is also shown (in blue).**





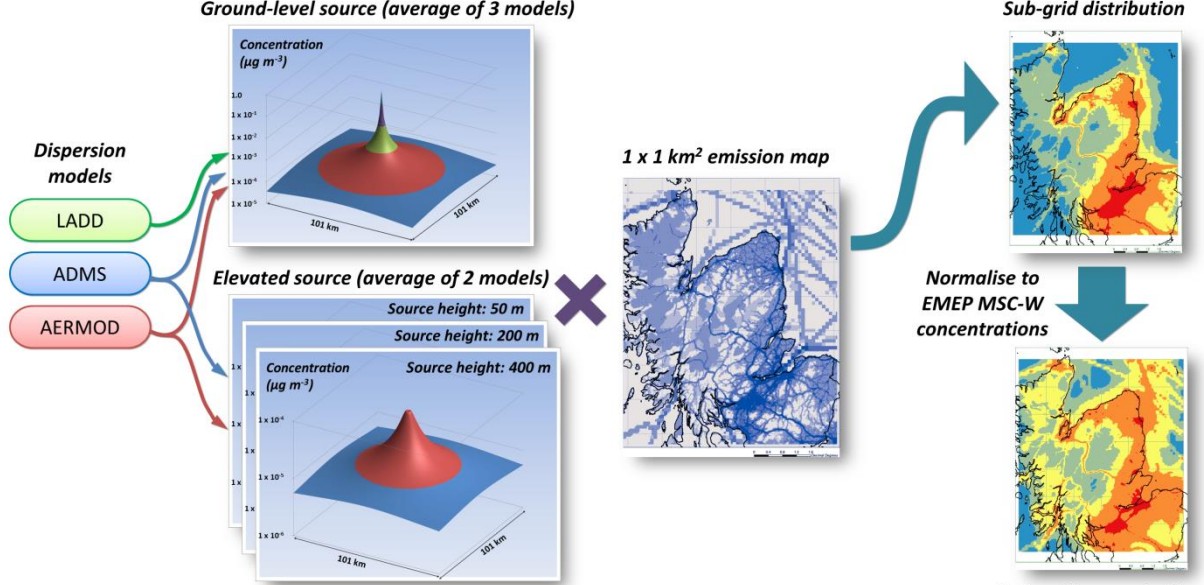

**Figure 2: Schematic showing the process of producing the sub-grid concentration predictions from short-range dispersion model simulations and high spatial resolution emission data.**





**Figure 3: Sub-grid model predictions (top row) of annual mean concentrations of NO$_2$ and NH$_3$ for the two domains. EMEP model predictions at a resolution of $50 \times 50$ km$^2$ are shown for comparison (bottom row).**



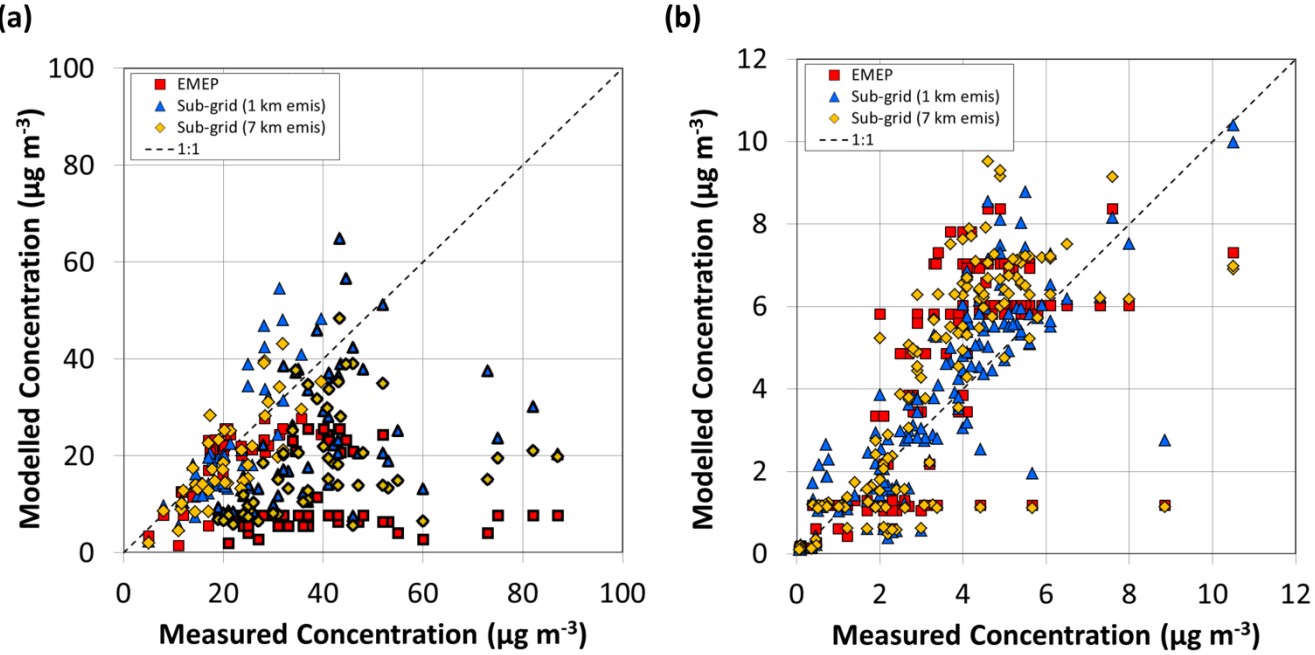

**Figure 4: Modelled concentrations plotted against measured values for all sites for (a) NO₂ and (b) NH₃. NO₂ traffic stations are indicated by bold symbol outlines. Plot data provided in the supplementary material.**

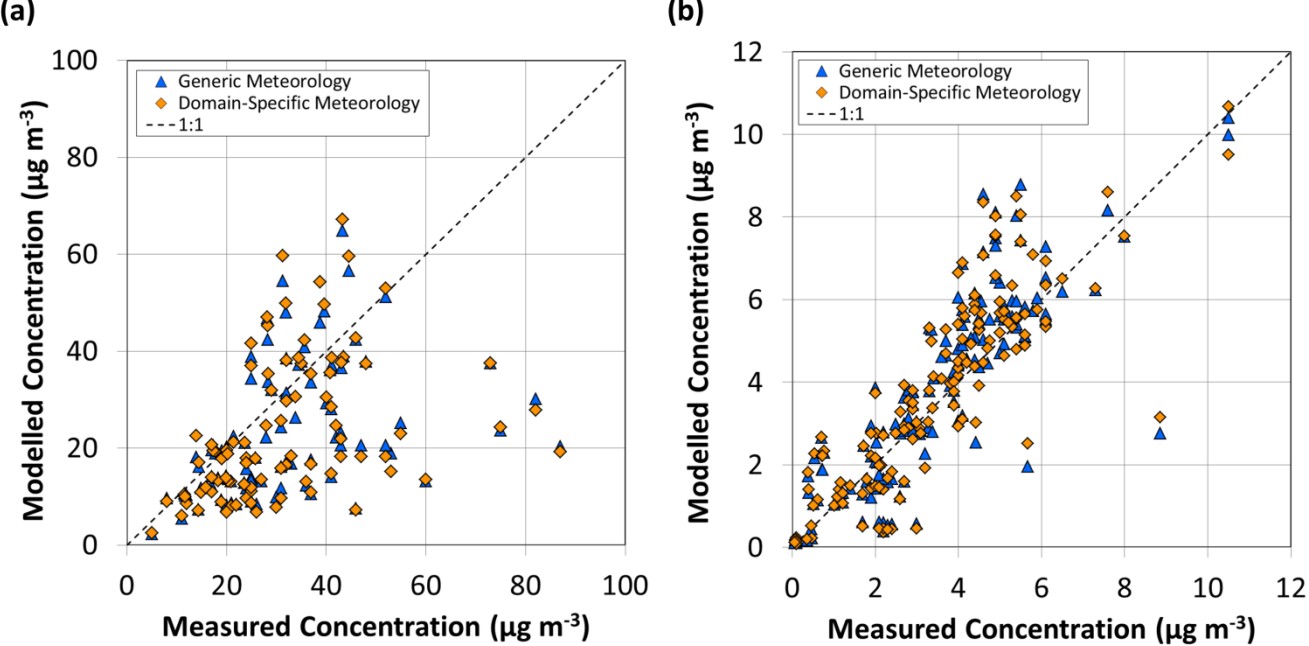

5   **Figure 5: Modelled concentrations plotted against measured values for all sites for (a) NO₂ and (b) NH₃ using both the generic meteorological dataset (LYNE95mod) and the domain-specific meteorological data. Plot data provided in the supplementary material.**