# Peer review of "A sub-grid model for improving the spatial resolution of air quality modelling at a European scale"

_Geoscientific Model Development, 2016_

## Short Comment (SC1) · 20 Jul 2016

Dear authors,

In my role as Executive editor of GMD, I would like to bring to your attention our Editorial version 1.1:

http://www.geosci-model-dev.net/8/3487/2015/gmd-8-3487-2015.html

This highlights some requirements of papers published in GMD, which is also available on the GMD website in the 'Manuscript Types' section:

http://www.geoscientific-model-development.net/submission/manuscript_types.html

In particular, please note that for your paper, the following requirements have not been met in the Discussions paper:

- "The main paper must give the model name and version number (or other unique identifier) in the title."

- "If the model development relates to a single model then the model name and the version number must be included in the title of the paper. If the main intention of an article is to make a general (i.e. model independent) statement about the usefulness of a new development, but the usefulness is shown with the help of one specific model, the model name and version number must be stated in the title. The title could have a form such as, "Title outlining amazing generic advance: a case study with Model XXX (version Y)"."

In order to simplify reference to your developments, please add a model name and a version number in the title of your article in your revised submission to GMD.

Yours,

Astrid Kerkweg

---

## Author Comment (AC1) · 20 Jul 2016

In response to the short comment by the Executive editor, Astrid Kerkweg, a model name and a version number will be included in the revised submission.

---

## Referee Comment (RC1) · Anonymous Referee #1 · 12 Aug 2016

General Comments

This paper tackles the issue of insufficient spatial resolution of modeled pollutant concentrations over Europe. I think that this is an open issue and therefore, new research in this area is definitely welcome. The authors present a post-processing technique, that combines finely resolved emission maps and dispersion model simulations to downscale regional chemistry-transport simulations at finer scale. I appreciate the special effort that has been made to keep the application simple and generalizable to different case-studies. However, I think that with the simplifications the authors made in order to keep their application 'light' – namely the use of a meteorology that does not match the case study and the averaging over annual time scale – they shift the

original question from the actual scientific issue of the unresolved sub grid scale variability to the very technical one "how to deal with European air-quality modeling since no high resolution inventory is available all over the continent". I wonder if the developed methodology is of any interest over areas where such high resolution bottom-up inventories already exist (e.g. the US), or in a future where such an inventory becomes available over Europe.

What strikes me most with the manuscript is the use of the term 'subgrid model'. The authors claim to develop a subgrid model to simulate pollutant concentration variability within regional CTM grid cells, but in my view, what they develop is a post-processing, downscaling technique to map regional scale simulations on finer resolution emission data using some parametrization to account for dispersion of these emissions. I think that referring to this technique as 'modeling the sub grid scale variability' is misleading because there is nothing in the formulation of the CTM that has been changed here to actually model the unresolved variability. If their model was actually a sub grid scale model I think that the authors should have refer in the introduction to the different approaches developed so far: Galmarini et al., 2008, Cassini et al., 2010, Korsakissok and Mallet 2010, Valari and Menut 2010 among others. Their references though, span the field of post-processing downscaling techniques rather than the issue of solving the unresolved sub grid variability.

Having said so, I think that seen as a post-processing, geostastical, downscaling technique that combines a fine resolution emission map as proxy with some assumptions on pollutant dispersion, the research presented in the manuscript is worth publishing. This would require significant changes in the formulations starting from the title and abstract. The introduction, discussion and conclusions should also change to put the work on this different framework.

Specific comments

For simplicity and to make their application easily reproduced, the authors chose to use

a generic meteorology, not simultaneous with their case-study. The parametrization of emission dispersion does not include any sub grid process such as fast chemistry or deposition. What is more, the time scale of their application is much coarser than the CTM's (annual vs. several minutes though the time step of the EMEP model is not explicitly mentioned). Those choices, make it clear, in my opinion, that this effort is not meant to solve the subgrid variability. To do so, the effort would rather focus on high resolution meteorology and emissions on line with the regional CTM to capture the unresolved features of atmospheric chemistry and dispersion.

p.1 ln 25: I am not convinced that this method could be extended in shorter time scales and other pollutants. The meteorology does not match the case-study and I think that looking at hourly or even daily data the discrepancies due to this mismatch should become very large. Since there is no coupling with chemistry, this approach seems to me relevant only for chemically inert species (or at time scales where active species could be considered as inert).

p2. ln 20-25: If the mean value is correct it would be surprising that the urban background concentration is underestimated. It would make sense to say that near-sources concentration levels are under predicted but if the background value is off as well, I don't see how we could get the mean value right.

p4. ln 5: Wouldn't it make more sense to use the same meteorology as in the EMEP model at least for this sensitivity test?

p.5 ln 30: I am wondering what do emission sources as large as 1km2 could possibly represent. In my understanding, dispersion models are conceived to represent emission from point sources such as industrial stacks. Is this the right model to represent large area sources such as crops or residential emissions? Is this type of modeling adequate to represent dispersion around busy roads? Don't dispersion patterns depend on the emission sector?

p7 ln 10: I don't think it is appropriate to say that "the sub grid model preformed better

than the EMEP model". It would be more fair to say that the downscaled version of the EMEP model compares better with observations. The same remark applies on many formulations throughout the results section.

p7 ln15: This makes me wonder how would results look like if no dispersion was taken into account and the same process was done only by using the 1km (or 7km or both) emission proxy.

p8 ln 5-10: I think it would be interesting to look at the effect of the meteorological dataset at a finer time-scale. Especially since the authors claim in their conclusions that this method is easily applicable at finer time scales.

p9 ln 20-25: I think that the comparison with Denby et al., 2011 study is off mainly because they worked on hourly data and not annual.

p9 ln 30: I think that the correlation in Schaap et al., 2015 is on time and not in space as in the present study.

p10 ln 10: The question inevitable arises of whether a direct EMEP run at 7km resolution with its corresponding meteorology would bring about the same improvement as the downscaling developed in the present study. And in this case the data would be directly at hourly resolution.

Please also note the supplement to this comment:
http://www.geosci-model-dev-discuss.net/gmd-2016-160/gmd-2016-160-RC1-supplement.pdf

---

## Referee Comment (RC2) · Anonymous Referee #2 · 23 Aug 2016

**1   General Comments**

This is an interesting paper describing a downscaling approach to model sub-grid variability of NO2 and NH3 concentrations at a resolution of 1km, starting from a regional scale CTM with 50km grid size. This is undoubtedly an important topic, as air pollution impact indicators such as population exposure to NO2 or PM show strong local variations with source strength, so that a grid size of 50km is clearly too coarse for impact assessment.

Although the paper is well written, I have a few comments that in my opinion need to be addressed before publication. I have doubts about the use of non-local meteorological

data that are inconsistent with the coarse CTM, and about the missing chemistry in the downscaling scheme when addressing reactive gases like NO2. Somehow there is an inherent mismatch between the aim to "resolve the large horizontal gradients found close to sources of relatively short-lived pollutants" (p.2 l 18) and the use of an approach that explicitly does not include photochemistry, formation of secondary aerosol or deposition processes. This method would be perfectly fine if the focus was on chemically relatively inert species like PM (particularly primary) or NOx. The methodology needs to be better justified there (if not improved) and the potential implications discussed.

**2 Specific comments**

P2 l20. One issue that is not addressed here nor anywhere else in the manuscript is that (to my understanding) the lowest vertical layer of the EMEP model extends from the surface to about 90 meters. In this context, it would be important to explain what is meant by 'the mean atmospheric concentration in each grid square': Is this the estimated surface concentration calculated by applying some standard vertical distribution? But if so, is it then justified to assume that the modelled mean concentration on the 50km grid is 'correct' and just needs to be re-distributed spatially within the grid cell?

P4 l29ff. I find it hard to understand why meteorological data from just one station was used here, instead of grid-specific meteorology consistent with the fields driving the CTM. Even the two "domain-specific" meteorological data sets come only from one station each. I think this needs to be better justified and compared to the effects a grid specific met data set would have.

P5 l16. I agree absolutely. I would not expect a 1x1km grid cell to be representative of a hotspot traffic station such as the high-concentration cases in Scotland. I don't see

how these could be resolved without a proper street canyon increment calculation. In this light, some of the discussion of the model performance later on (p9 l3 ff) could be a bit rephrased.

p.5l 27 – whole Section 3: I would find it helpful if the downscaling equations were written out as equations and not only described verbally. I do appreciate the process flowchart in the supplementary material which is very helpful in understanding the procedure and should definitely be kept, but I believe also a formal treatment could help here.

p.6 l1. I have difficulties understanding why NO2 photochemistry is not needed here. By using the NOx fine-scale emission pattern and distribution kernel for redistributing NO2 concentrations this approach assumes a constant NO2/NOx concentration ratio in the whole EMEP 50km grid cell, which is hard for me to believe. Typically this ratio should show a dip close to NOx sources due to the high NO ratio in primary emissions. Also this fraction of primary NO2 in NOx emissions varies strongly between different sources and this should play a role at the local scale. I thought that at least ADMS includes a NOx photochemistry formulation, why is this not used? Ideally such a chemistry scheme should be used, or at least the potential errors discussed. Along similar lines, also the formation of NH4 can influence local NH3 concentrations - it would be interesting to know what the potential errors are when these processes are ignored.

p6 l19. This is a part of the model formulation I have not understood. Why is it justified to assume rotationally symmetric dispersion kernels, should not the local predominant wind direction have a significant influence? How big is the error introduced by this rotationally symmetric formulation?

P6 l24 Why moving window? Is the final subgrid distribution pattern at a given 1km grid cell not simply made up from the sum of all emissions times kernel within the range of the kernel? Perhaps a better formulation could be to say that the kernels are not cut

Interactive
comment

off at the EMEP grid boundaries (I hope). Again, formulating this with equations would help here.

P7 l16. Does this mean the downscaling decreased agreement to NO2 non-traffic stations, which are the ones we would actually expect it to improve? As mentioned above, I would not expect the 1x1km model to be representative of traffic stations.

P10 l17. In case of PM10 to my knowledge the limit on daily means (35 exceedances of 50ug/m3) is the more stringent one.

P10 l22. Since the authors mention this as a possibility I would suggest that they add a sensitivity analysis using grid-specific meteorology consistent with the CTM.

P10 l31. . . .for other pollutants for which there is a strong relationship between emissions and concentrations (i.e. which have very localized sources) and – in my opinion – which do not undergo fast chemical transformation. Therefore I agree in the case of PM2.5, especially primary PM, but I have doubts for NO, SO4 and O3.

**3 Technical corrections**

p6 l24-26. I wonder whether the quotes are really necessary here.

P11 l2. $SO_2^{4-}$ should be $SO_4^{2-}$?

Figure 4: I would suggest to split up the two regions into separate figures, since the model performance is so different.
* * *

---

## Author Comment (AC2) · 31 Aug 2016

We would first like to thank both referees for their useful and constructive comments. In order to stimulate discussion between the referees and the authors we have combined the response to both Anonymous Referees. The responses have been grouped by topic to improve readability.

Approach used

Ref#1: Referring to this technique as 'modeling the sub grid scale variability' is misleading because there is nothing in the formulation of the CTM that has been changed here to actually model the unresolved variability. In my view, what they develop is

a post-processing, downscaling technique to map regional scale simulations on finer resolution emission data using some parametrization to account for dispersion of these emissions.

Ref#1: The parametrization of emission dispersion does not include any sub grid process such as fast chemistry or deposition. What is more, the time scale of their application is much coarser than the CTM's (annual vs. several minutes though the time step of the EMEP model is not explicitly mentioned). Those choices make it clear, in my opinion, that this effort is not meant to solve the subgrid variability. To do so, the effort would rather focus on high resolution meteorology and emissions on line with the regional CTM to capture the unresolved features of atmospheric chemistry and dispersion.

Authors' response: We appreciate the referee's opinion on this matter. It is a question of definitions. The referee's interpretation of the term "sub-grid model" appears to be that of a parameterisation within a coarse resolution model (e.g. a CTM) that explicitly takes into account sub-grid variability during the simulation. We use the term "sub-grid model" to mean a parameterisation that can refine the outputs of a coarse resolution model, by providing an estimate of the spatial distribution of the model output (e.g. atmospheric concentrations) within each grid cell of the coarser model. In this respect we a using the term "sub-grid model" in the same sense as that used by various publications cited in the manuscript (Ching et al., 2006; Denby et al., 2011; Dragosits et al., 2002 and Isakov et al., 2007). The terms 'sub-grid model' and 'sub-grid variability' will be defined in the revised manuscript to avoid confusion.

Ref#2: I have difficulties understanding why NO2 photochemistry is not needed here. By using the NOx fine-scale emission pattern and distribution kernel for redistributing NO2 concentrations this approach assumes a constant NO2/NOx concentration ratio in the whole EMEP 50km grid cell, which is hard for me to believe. Typically this ratio should show a dip close to NOx sources due to the high NO ratio in primary emissions. Also this fraction of primary NO2 in NOx emissions varies strongly between

different sources and this should play a role at the local scale. I thought that at least ADMS includes a NOx photochemistry formulation, why is this not used? Ideally such a chemistry scheme should be used, or at least the potential errors discussed. Along similar lines, also the formation of NH4 can influence local NH3 concentrations – it would be interesting to know what the potential errors are when these processes are ignored.

Authors' response: The assumption of a constant NO2/NOx concentration ratio is made to provide a simple parameterisation that is universally applicable. It would be difficult to include a variable ratio in a simple model since it would depend on the local pollution climate (e.g. ozone concentrations), photochemical reactions of NO and NO2 emitted from the different source types and complex variations in diurnal emission patterns and meteorological drivers. Both AERMOD and ADMS include optional simple photochemistry schemes but they depend on background ozone concentrations, which are not homogeneous across the domains, and so these model options cannot be used for developing a simple generic model, which was the aim of the study. Of course this simplification will introduce uncertainty into the model. This uncertainty can be estimated by analysing the variability of the NO2/NOx ratio in the measured data. For the Scottish domain, the coefficient of variation of the ratio is 20% whereas for the Dutch domain it is only 7.5%. Estimating the NO2 concentrations from the measured NOx concentrations, assuming a constant ratio, gives mean errors (NMGE) of 18% and 6.3%, for the Scottish and Dutch domains, respectively. Extending this analysis to the annual mean concentrations for the sites in the European air pollution database "airbase" that simultaneously measure NO and NO2 (1478 sites), gives a coefficient of variation of 17.5% and a mean error of 14.6%. Similarly the formation of NH4+ from NH3 depends on the concentrations of other pollutants, as well as the complex variations in diurnal emission patterns and meteorological drivers, the inclusion of which is beyond the scope of a simple sub-grid model. A discussion of these uncertainties will be included in the revised manuscript

Ref#1: How would results look like if no dispersion was taken into account and the same process was done only by using the 1km (or 7km or both) emission proxy?

Authors' response: This is an interesting question and should be fairly easy to test. We will include this additional analysis in the revised manuscript

Meteorological data

Ref#2: I find it hard to understand why meteorological data from just one station was used here, instead of grid-specific meteorology consistent with the fields driving the CTM. Even the two "domain-specific" meteorological data sets come only from one station each. I think this needs to be better justified and compared to the effects a grid specific met data set would have.

Ref#2: Since the authors mention this as a possibility I would suggest that they add a sensitivity analysis using grid-specific meteorology consistent with the CTM

Ref#1: Wouldn't it make more sense to use the same meteorology as in the EMEP model at least for this sensitivity test?

Authors' response: The model was developed using a single meteorological dataset to provide a simple parameterisation that could be of benefit to the wider air quality modelling community. The model was tested with meteorology specific to each modelling domain to show that the model results were not that sensitive to the meteorological data set used. The use of grid-specific met data would require nearly 800 ADMS and AERMOD simulations (110 grid cells for 7 emission heights) and 110 LADD simulations (one for each grid cell). However, before revising the manuscript we will assess the feasibility of carrying out this additional analysis.

Ref#2: Why is it justified to assume rotationally symmetric dispersion kernels, should not the local predominant wind direction have a significant influence? How big is the error introduced by this rotationally symmetric formulation?

Authors' response: In order to keep the model parameterisation simple and universally

applicable, we decided to remove the influence of local wind direction distributions (which are not valid for the whole domain) by assuming rotationally symmetric dispersion kernels. It can anyway be noted that in the real world local wind-directions will differ from that of a larger scale model. For example, cities are frequently located in locations subject to topographic (e.g. valley) winds, or sea-breezes. Although this rotation symmetry obviously does introduce some error, the results shown in Figs. 4 and 5 demonstrate that even with this assumption the sub-grid model provides a valuable improvement over that of the larger scale CTM. This assumption will be discussed in the revised manuscript.

Emissions

Ref#1: I am wondering what do emission sources as large as 1km2 could possibly represent. In my understanding, dispersion models are conceived to represent emission from point sources such as industrial stacks. Is this the right model to represent large area sources such as crops or residential emissions? Is this type of modeling adequate to represent dispersion around busy roads? Don't dispersion patterns depend on the emission sector?

Authors' response: The use of 1 km2 emission sources is a simplification due to the fact that we do not have emission inventories for the study domains at a higher spatial resolution. So in reality we may have a large point source within a specific 1 x 1 km2 grid cell but since we do not know exactly where, we distribute this emission over the whole grid cell. Of course this is a simplification which is going to introduce uncertainty in the model but it is the best we can do with the available emission data. The model, therefore, would not be expected to be suitable for estimating concentrations close to busy roads but the model performs well for the Dutch traffic sites. This will be made clearer in the revised manuscript.

Averaging period

Ref#1: It would be interesting to look at the effect of the meteorological dataset at a

finer time-scale. Especially since the authors claim in their conclusions that this method is easily applicable at finer time scales.

Authors' response: As stated in the manuscript, the annual mean limit values for NO2 are generally more stringent than the hourly ones and impacts of NH3 are only assessed using annual mean concentrations. We agree that it would be interesting to develop a model with a higher temporal resolution, but such a study would raise other issues (e.g. as discussed above, or with the availability and accuracy of time-resolved emissions). Thus, we did not consider this to be 'easily applicable' at finer time-scales, but rather, as we state in the discussion, this is a potential future improvement and is out of the scope of the presented work.

Model evaluation

Ref#1: I don't think it is appropriate to say that "the sub grid model performed better than the EMEP model". It would be more fair to say that the downscaled version of the EMEP model compares better with observations.

Authors' response: This is a fair point and the manuscript will be modified to reflect this.

Ref#2: Does this mean the downscaling decreased agreement to NO2 non-traffic stations, which are the ones we would actually expect it to improve? As mentioned above, I would not expect the 1x1km model to be representative of traffic stations.

Authors' response: You would expect the sub-grid model to perform best for the non-traffic stations but for the Dutch domain it had the lowest bias and error for the traffic sites. This suggests that the model may be suitable for simulating concentrations at traffic stations, providing they are located several metres from the roadside (as is the case for most of the Dutch sites), and the good agreement is likely the result of the fact that the majority of roadside sites are not situated along isolated motorways, but rather embedded in urban areas with dense road networks. A discussion of this will be

included in the revised manuscript.

Manuscript structure

Ref#2: I would find it helpful if the downscaling equations were written out as equations and not only described verbally

Authors' response: This is a good suggestion and the downscaling equations will be included in the manuscript.

Miscellaneous comments

Ref#1: The inevitable question arises of whether a direct EMEP run at 7km resolution with its corresponding meteorology would bring about the same improvement as the downscaling developed in the present study. And in this case the data would be directly at hourly resolution.

Authors' response: Although it is currently possible to run the EMEP model at a 7 km resolution for specific projects, they are far too CPU intensive for routine use, and especially where CTMs need to be run 10s-100s of time for emission control assessments. However even at a spatial resolution of 7 km, the EMEP model would not be able to resolve the large horizontal concentration gradients found close to sources and a sub-grid treatment may still be necessary. A discussion of this will be included in the revised manuscript.

Ref#2: One issue that is not addressed here nor anywhere else in the manuscript is that (to my understanding) the lowest vertical layer of the EMEP model extends from the surface to about 90 meters. In this context, it would be important to explain what is meant by 'the mean atmospheric concentration in each grid square': Is this the estimated surface concentration calculated by applying some standard vertical distribution? But if so, is it then justified to assume that the modelled mean concentration on the 50km grid is 'correct' and just needs to be re-distributed spatially within the grid cell?

Authors' response: The EMEP model estimates the near surface concentrations by extrapolating the concentration at the mid height of the first vertical layer (45 m) assuming an approximately constant vertical deposition flux. Of course, there are uncertainties in this procedure, especially for NH3 where deposition velocities can be relatively high, but where bi-directional exchange complicates even the sign of atmosphere-biosphere exchange in real situations. However, we have to work with the EMEP values as provided by the model, and our working assumption is that the errors in the vertical distribution are less important than the errors we seek to minimise – namely those of sub-grid horizontal variability. This will be clarified in the revised manuscript.

Please also note the supplement to this comment:
http://www.geosci-model-dev-discuss.net/gmd-2016-160/gmd-2016-160-AC2-supplement.pdf

---

## Author Comment (AC3) · 25 Oct 2016

Updated Authors' response to the comments by Anonymous Referees #1 and #2

We would first like to thank both referees for their useful and constructive comments. We have combined the responses to both Anonymous Referees and the responses have been grouped by topic to improve readability. Page and line numbers refer to the manuscript version with Tracked Changes (provided as a supplement to these comments).

Approach used

Ref#1: Referring to this technique as 'modeling the sub grid scale variability' is mis-

leading because there is nothing in the formulation of the CTM that has been changed here to actually model the unresolved variability. In my view, what they develop is a post-processing, downscaling technique to map regional scale simulations on finer resolution emission data using some parametrization to account for dispersion of these emissions.

Ref#1: The parametrization of emission dispersion does not include any sub grid process such as fast chemistry or deposition. What is more, the time scale of their application is much coarser than the CTM's (annual vs. several minutes though the time step of the EMEP model is not explicitly mentioned). Those choices make it clear, in my opinion, that this effort is not meant to solve the subgrid variability. To do so, the effort would rather focus on high resolution meteorology and emissions on line with the regional CTM to capture the unresolved features of atmospheric chemistry and dispersion.

Authors' response: We appreciate the referee's opinion on this matter. We think that it is a question of definitions. The referee's interpretation of the term "sub-grid model" appears to be that of a parameterisation within a coarse resolution model (e.g. a CTM) that explicitly takes into account sub-grid variability during the simulation. We use the term "sub-grid model" to mean a parameterisation that can refine the outputs of a coarse resolution model, by providing an estimate of the spatial distribution of the model output (e.g. atmospheric concentrations) within each grid cell of the coarser model. In this respect we a using the term "sub-grid model" in the same sense as that used by various publications cited in the manuscript (Ching et al., 2006; Denby et al., 2011; Dragosits et al., 2002 and Isakov et al., 2007). The manuscript has been revised to clarify what we mean by sub-grid modelling (see e.g. p 4 lines 13-14).

Ref#2: I have difficulties understanding why NO2 photochemistry is not needed here. By using the NOx fine-scale emission pattern and distribution kernel for redistributing NO2 concentrations this approach assumes a constant NO2/NOx concentration ratio in the whole EMEP 50km grid cell, which is hard for me to believe. Typically this

ratio should show a dip close to NOx sources due to the high NO ratio in primary emissions. Also this fraction of primary NO2 in NOx emissions varies strongly between different sources and this should play a role at the local scale. I thought that at least ADMS includes a NOx photochemistry formulation, why is this not used? Ideally such a chemistry scheme should be used, or at least the potential errors discussed. Along similar lines, also the formation of NH4 can influence local NH3 concentrations – it would be interesting to know what the potential errors are when these processes are ignored.

Authors' response: The assumption of a constant NO2/NOx concentration ratio is made to provide a simple parameterisation that is universally applicable. It would be difficult to include a variable ratio in a simple model since it would depend on the local pollution climate (e.g. ozone concentrations), photochemical reactions of NO and NO2 emitted from the different source types and complex variations in diurnal emission patterns and meteorological drivers. Both AERMOD and ADMS include optional simple photochemistry schemes but they depend on background ozone concentrations, which are not homogeneous across the domains, and so these model options cannot be used for developing a simple generic model, which was the aim of the study.

Of course this simplification will introduce uncertainty into the model. This uncertainty can be estimated by analysing the variability of the NO2/NOx ratio in the measured data. For the Scottish domain, the coefficient of variation of the ratio is 20% whereas for the Dutch domain it is only 7.5%. Estimating the NO2 concentrations from the measured NOx concentrations, assuming a constant ratio, gives mean errors (NMGE) of 18% and 6.3%, for the Scottish and Dutch domains, respectively. Extending this analysis to the annual mean concentrations for the sites in the European air pollution database "airbase" that simultaneously measure NO and NO2 (1478 sites), gives a coefficient of variation of 17.5% and a mean error of 14.6%. Based on the European dataset, we estimate that the mean error introduced by assuming a constant NO2/NOx ratio is about 16%. A discussion of these uncertainties has been included in the Discussion section (p 12 lines 23-28).

Similarly the formation of NH4+ from NH3 depends on the concentrations of other pollutants, as well as the complex variations in diurnal emission patterns and meteorological drivers, the inclusion of which is beyond the scope of a simple sub-grid model. However, a discussion of the uncertainty due to this omission has been included in the Discussion section (p 12 line 28 to p 13 line 1).

Ref#1: How would results look like if no dispersion was taken into account and the same process was done only by using the 1km (or 7km or both) emission proxy?

Authors' response: We have now included these analyses using the 1 x 1 km2 emission data for both NO2 and NH3 (p 10 lines 6-15 and Figure 6). As now mentioned in the discussion, these results show that short-range dispersion estimates are necessary for improving on the EMEP model predictions.

Meteorological data

Ref#2: I find it hard to understand why meteorological data from just one station was used here, instead of grid-specific meteorology consistent with the fields driving the CTM. Even the two "domain-specific" meteorological data sets come only from one station each. I think this needs to be better justified and compared to the effects a grid specific met data set would have.

Ref#2: Since the authors mention this as a possibility I would suggest that they add a sensitivity analysis using grid-specific meteorology consistent with the CTM

Ref#1: Wouldn't it make more sense to use the same meteorology as in the EMEP model at least for this sensitivity test?

Authors' response: The model was developed using a single meteorological dataset to provide a simple parameterisation that could be of benefit to the wider air quality modelling community. The model was tested with meteorology specific to each modelling domain to show that the model results were not that sensitive to the meteorological

data set used. However, we agree that it makes more sense to use the same meteorology as the EMEP model and we have done this for the revised manuscript. The original approach using the modified meteorological data from just one station is now included as a sensitivity test and is now referred to as the simulation using "synthetic meteorological data". As can be seen in Figure 5, the use of the EMEP model meteorology does not make a large difference to the concentration predictions and model performance is similar, strengthening the conclusion that the approach is fairly insensitive to the meteorological dataset used. The manuscript has been revised to reflect this change in methodology (see e.g. p5 lines 5-12; p 7 lines 7-8 and lines 17-19).

Ref#2: Why is it justified to assume rotationally symmetric dispersion kernels, should not the local predominant wind direction have a significant influence? How big is the error introduced by this rotationally symmetric formulation?

Authors' response: In order to keep the model parameterisation simple and universally applicable, we decided to remove the influence of local wind direction distributions (which are not valid for the whole domain) by assuming rotationally symmetric dispersion kernels. It can anyway be noted that in the real world local wind-directions will differ from that of a larger scale model. For example, cities are frequently located in locations subject to topographic (e.g. valley) winds, or sea-breezes. Although this rotation symmetry obviously does introduce some error, the results shown in Figs. 4 and 5 demonstrate that even with this assumption the sub-grid model provides a valuable improvement over that of the larger scale CTM. The use of EMEP model meteorology, however, now relegates this approach to that of a sensitivity test.

Emissions

Ref#1: I am wondering what do emission sources as large as 1km2 could possibly represent. In my understanding, dispersion models are conceived to represent emission from point sources such as industrial stacks. Is this the right model to represent large area sources such as crops or residential emissions? Is this type of modeling adequate

to represent dispersion around busy roads? Don't dispersion patterns depend on the emission sector?

Authors' response: The use of 1 km2 emission sources is a simplification due to the fact that we do not have emission inventories for the study domains at a higher spatial resolution. So in reality we may have a large point source within a specific 1 x 1 km2 grid cell but since we do not know exactly where, we distribute this emission over the whole grid cell. Of course this is a simplification which is going to introduce uncertainty in the model but it is the best we can do with the available emission data. The model, therefore, would not be expected to be suitable for estimating concentrations close to busy roads but the model performs well for the Dutch traffic sites. This has been made clearer in the revised manuscript (p 13 lines 5-11).

Averaging period

Ref#1: It would be interesting to look at the effect of the meteorological dataset at a finer time-scale. Especially since the authors claim in their conclusions that this method is easily applicable at finer time scales.

Authors' response: As stated in the manuscript, the annual mean limit values for NO2 are generally more stringent than the hourly ones and impacts of NH3 are only assessed using annual mean concentrations. We agree that it would be interesting to develop a model with a higher temporal resolution, but such a study would raise other issues (e.g. as discussed above, or with the availability and accuracy of time-resolved emissions). Thus, we did not consider this to be 'easily applicable' at finer time-scales, but rather, as we state in the discussion, this is a potential future improvement and is out of the scope of the presented work.

Model evaluation

Ref#1: I don't think it is appropriate to say that "the sub grid model performed better than the EMEP model". It would be more fair to say that the downscaled version of the

EMEP model compares better with observations.

Authors' response: This is a fair point and the manuscript has been modified accordingly.

Ref#2: Does this mean the downscaling decreased agreement to NO2 non-traffic stations, which are the ones we would actually expect it to improve? As mentioned above, I would not expect the 1x1km model to be representative of traffic stations.

Authors' response: You would expect the sub-grid model to perform best for the non-traffic stations but for the Dutch domain it had the lowest bias and error for the traffic sites. This suggests that the model may be suitable for simulating concentrations at traffic stations, providing they are located several metres from the roadside (as is the case for most of the Dutch sites), and the good agreement is likely the result of the fact that the majority of roadside sites are not situated along isolated motorways, but rather embedded in urban areas with dense road networks. This is now highlighted in the Discussion (p 13 lines 8-10).

Manuscript structure

Ref#2: I would find it helpful if the downscaling equations were written out as equations and not only described verbally

Ref#2: Why moving window? Is the final subgrid distribution pattern at a given 1km grid cell not simply made up from the sum of all emissions times kernel within the range of the kernel? Perhaps a better formulation could be to say that the kernels are not cut off at the EMEP grid boundaries (I hope). Again, formulating this with equations would help here.

Authors' response: This is a good suggestion and the downscaling equation is now included in the Model Development section. The dispersion kernels are not cut-off at the EMEP grid boundaries. This is now clarified in the description of the downscaling equation (p 7 lines 26-27).

[Figure]

Miscellaneous comments

Ref#1: The inevitable question arises of whether a direct EMEP run at 7km resolution with its corresponding meteorology would bring about the same improvement as the downscaling developed in the present study. And in this case the data would be directly at hourly resolution.

Authors' response: Although it is currently possible to run the EMEP model at a 7 km resolution for specific projects, they are far too CPU intensive for routine use, and especially where CTMs need to be run 10s-100s of time for emission control assessments. However even at a spatial resolution of 7 km, the EMEP model would not be able to resolve the large horizontal concentration gradients found close to sources and a sub-grid treatment may still be necessary. This is now clarified in the Introduction (p 3 lines 15-16).

Ref#2: One issue that is not addressed here nor anywhere else in the manuscript is that (to my understanding) the lowest vertical layer of the EMEP model extends from the surface to about 90 meters. In this context, it would be important to explain what is meant by 'the mean atmospheric concentration in each grid square': Is this the estimated surface concentration calculated by applying some standard vertical distribution? But if so, is it then justified to assume that the modelled mean concentration on the 50km grid is 'correct' and just needs to be re-distributed spatially within the grid cell?

Authors' response: The EMEP model estimates the near surface concentrations by extrapolating the concentration at the mid height of the first vertical layer (45 m) assuming an approximately constant vertical deposition flux. Of course, there are uncertainties in this procedure, especially for NH3 where deposition velocities can be relatively high, but where bi-directional exchange complicates even the sign of atmosphere-biosphere exchange in real situations. However, we have to work with the EMEP values as provided by the model, and our working assumption is that the errors in the vertical distribution are less important than the errors we seek to minimise – namely those of sub-grid horizontal variability. This is now clarified in the Introduction (p 2 lines 20-22).

Ref#1: If the mean value is correct it would be surprising that the urban background concentration is underestimated. It would make sense to say that near-sources concentration levels are under predicted but if the background value is off as well, I don't see how we could get the mean value right.

Authors' response: Although this comment refers to the conclusions stated by Denby et al. (2011), we will attempt to respond to the referee's comment. Both rural and urban areas can be present within a 50 km grid square of the EMEP model and so it is possible that the model predicts the correct mean value for the entire grid square but underestimates concentrations within the urban area (at both traffic and background locations). This would be the case, for example, if the rural area occupies the majority of the grid square.

Ref#1: If I think that the comparison with Denby et al., 2011 study is off mainly because they worked on hourly data and not annual.

Authors' response: We disagree with this comment because the study of Denby et al. (2011) is based on annual mean concentrations.

Ref#1: If I think that the correlation in Schaap et al., 2015 is on time and not in space as in the present study.

Authors' response: We are referring to Figure 7 of Schaap et al. (2015), which is included in the section on spatial analysis (although it appears to be erroneously referred to as "temporal analysis" in the figure caption).

Ref#2: In case of PM10 to my knowledge the limit on daily means (35 exceedances of 50ug/m3) is the more stringent one.

Authors' response: Yes the referee is correct, for PM10 the limit for daily means is the more stringent one. The reference to PM10 has been removed from the revised

manuscript (p 12 line 17)

Ref#2: I wonder whether the quotes are really necessary here.

Authors' response: Maybe not for the "moving window" but we think that the quotes necessarily highlight the new term "sub-grid distributions" in order to distinguish it from the term "sub-grid concentrations".

Ref#2: SO4-2 should be SO-24?

Authors' response: Yes, that is correct and the change has been made.

Ref#2: Figure 4: I would suggest to split up the two regions into separate figures, since the model performance is so different.

Authors' response: We agree and the figures for the individual domains have now been included in the supplementary material.

Additional changes to the manuscript

In addition to the changes mentioned above, the following additional modification has been made to the manuscript:

- The model name and version has been included in the manuscript title, as per journal guidelines and the model name and acronym used throughout the manuscript

Please also note the supplement to this comment:
http://www.geosci-model-dev-discuss.net/gmd-2016-160/gmd-2016-160-AC3-supplement.pdf

———————————————

[Figure]

**Supplement:**

**Improving the spatial resolution of air quality modelling at a European scale – Development and evaluation of the Air Quality Re-gridder Model (AQR v1.1)**

Mark R. Theobald[1,2], David Simpson[3,4], Massimo Vieno[5]

[revised manuscript text omitted]

Meteorological data used to develop the sub-grid model were taken

In order to parametrise the pollutant dispersion from source areas, three different atmospheric dispersion models were used. These were ADMS (v4.1) (Carruthers et al., 1994), AERMOD (v12345) (Cimorelli et al., 2002) and LADD (Dragosits et al., 2002). These three models were chosen because they have been extensively evaluated for the atmospheric dispersion of $NO_2$ and $NH_3$, with the exception of LADD, which has only been evaluated for $NH_3$ (Theobald et al., 2012). The meteorological data used for the atmospheric dispersion simulations were derived from the meteorological data used in the EMEP model simulation (generated by the Weather Research Forecast (WRF) model version 3.6.1; http://www.wrf-model.org). Surface and vertical profile data at the centre of each EMEP model grid square were extracted in AERMOD format using the Mesoscale Model Interface Program (MMIF; https://www3.epa.gov/ttn/scram/dispersion_related.htm#mmif) and subsequently converted into the input formats for ADMS and LADD. In order to test the sensitivity of AQR to the meteorological data used, additional simulations were carried out using two domain-specific real meteorological datasets and a synthetic meteorological dataset derived from data from an arbitrary location. The two domain-specific datasets used were from Easter Bush, for Scotland (von Bobrutzki et al., 2010), and Cabauw, for the Netherlands (obtained from the Cesar Database: http://www.cesar-database.nl/). The synthetic dataset was derived from data from the Lyneham meteorological station in the UK for 1995 (LYNE95) (Spanton et al., 2004), which was a fairly typical year with regards to mean air temperature and wind speed. Although an arbitrary choice, this dataset was chosen because it has been used in various model evaluation studies and has been made freely available to the dispersion modelling community (e.g. Hall et al., 2000; Theobald et al., 2012). In order to make theThis dataset less location specificwas modified (LYNE95mod) by randomising the wind direction data were randomised and scaling the wind speed was scaled so that the annual mean value was equal to the annual domain mean value used in the EMEP model for the 2008 study year (5.1 m s$^{-1}$). The use of a single UK meteorological dataset from a different year to the study year for the development of a model applied at the European scale may introduce a large amount of error or uncertainty in the predictions. In order to assess this uncertainty, two domain-specific meteorological datasets for the study year were also tested. These datasets were from Easter Bush, for Scotland (von Bobrutzki et al., 2010), and Cabauw, for the Netherlands (obtained from the Cesar Database: http://www.cesar-database.nl/).The wind directions were randomised for two reasons: 1) to make the meteorological data less location specific so that they can be used within different modelling domains and 2) to provide a generic dataset that could be of use to the air quality modelling community.

Evaluation of the AQR model

In order to parametrise the pollutant dispersion from source areas, three different atmospheric dispersion models were used. These were ADMS (v4) (Carruthers et al., 1994), AERMOD (v12345) (Cimorelli et al., 2002) and LADD (Dragosits et al., 2002). These three models were chosen because they have been extensively evaluated for the atmospheric dispersion of $NO_2$ and $NH_3$, with the exception of LADD, which has only been evaluated for $NH_3$ (Theobald et al., 2012).

Model evaluation was carried out using 2008 annual mean concentration data from local and national monitoring networks in the two study domains. For Scotland, $NO_2$, data were obtained from the Air Quality in Scotland website

(http://www.scottishairquality.co.uk/) (48 stations: 37 traffic and 11 non-traffic sites) and from RIVM for the Netherlands (43 stations: 13 traffic and 30 non-traffic). The evaluation was done for all sites, and for the traffic and non-traffic sites separately since the traffic sites are strongly influenced by the exact site location and are unlikely to be representative of a $1 \times 1$ km$^2$ grid square. For NH$_3$ concentrations in the Scottish domain, monitoring data were obtained from the UK National Ammonia Monitoring Network (NAMN) (Sutton et al., 2001) (http://uk-air.defra.gov.uk/networks/network-info?view=nh3), which has 14 sites within the domain. In addition, NH$_3$ monitoring data from 21 sites in a local network covering 36 km$^2$ (Vogt et al., 2013) were also used. For the Netherlands, NH$_3$ concentration data from the Measuring Ammonia in Nature (MAN) network (Lolkema et al., 2015) were provided by RIVM (108 stations). Model performance was assessed using the evaluation statistics of the R package "OpenAir" (Carslaw and Ropkins, 2012). Four performance metrics were used to compare the modelled concentrations with the observed values: fraction of model predictions within a factor of two of the observations (FAC2), normalised mean bias (NMB), normalised mean gross error (NMGE) and the Pearson correlation coefficient (r) (see Appendix A for definitions).

**3 Model development**

The sub-grid $1 \times 1$ km$^2$ concentration estimates were calculated from three components: the EMEP $50 \times 50$ km$^2$ concentration predictions, the $1 \times 1$ km$^2$ emission data and an estimate of short-range (< 50 km) pollutant dispersion. Figure 2 shows a schematic of the process. Short-range pollutant dispersion was parameterised using a simple scenario of a single $1 \times 1$ km$^2$ source with an emission rate of 1 Mg km$^{-2}$ yr$^{-1}$ in the centre of a square domain (of dimensions $101 \times 101$ km$^2$). Although individual sources are generally smaller than this, this value was used to match the spatial resolution of the emission data. For NO$_2$, the assumption was made that annual mean NO$_2$ concentrations are linearly correlated with those of NO$_x$. This allowed us to use the NO$_x$ emissions for the calculation of NO$_2$ concentrations without considering photochemical reactions. An analysis of the 2008 mean annual concentrations for the 1478 sites in the Air Quality e-Reporting database (formerly AirBase) of the European Environment Agency shows that measured NO$_2$ and NO$_x$ concentrations are approximately linearly correlated with a linear correlation coefficient, $r^2$ of 0.93. For the dispersion of NH$_3$, the source was assumed to be at ground level (a suitable approximation for most agricultural sources, which account for more than 90% of emissions in Europe). For NO$_2$, the assumption was made that NO$_2$ concentrations were linearly correlated with NO$_x$ concentrations (valid for annual mean concentrations). Emissions of NO$_x$, on the other hand, This allowed us to use the NO$_x$ emissions for the calculation of NO$_2$ concentrations without considering photochemical reactions. Emissions of NO$_x$ can occur over a range of emission heights, depending on the source type. Since the emission height will affect the resulting NO$_2$ concentrations at ground level, it needs to be taken into account. This was done by assigning a representative emission height for each emission sector (Selected Nomenclature for Air Pollution (SNAP) code) that contributed more than 1% of the total domain emissions (Table 1). These emission heights correspond loosely approximately to the mean effective emission heights used in the EMEP model for the sector emissions. In order to test the sensitivity of the sub-grid AQR model to the

emission heights used, additional simulations were carried out using emission heights half and double these values. For the ground level source, all three dispersion models (ADMS, AERMOD and LADD) were used to simulate the annual mean near-ground-level concentrations of $NH_3$ and $NO_2$ on a 1 km grid (for the $101 \times 101$ km$^2$ domain) using the LYNE95mod meteorological dataset.). For the elevated source scenarios, only ADMS and AERMOD were used to simulate the annual mean concentrations because the LADD model is not suitable for simulating dispersion from elevated sources (Theobald et al., 2012). A height of 1.5 m was used for the near-ground-level concentrations, because this height is commonly used for concentration monitoring and impact assessments (Cape et al., 2009). These short-range dispersion simulations were carried out using the meteorological data extracted from the WRF simulations at the centre of each EMEP model grid square. No removal processes (chemical reactions, dry or wet deposition etc.) were simulated because these processes depend strongly on local conditions (concentrations of other chemical species, meteorological conditions, surface characteristics, etc.).

The result of these simulations was nine concentration fields, (kernels), three for ground level sources (three models × one source height) and six for elevated sources (two models × three source heights) centred on the source location. For each source height, a rotationally symmetric concentration field (or kernel) was obtained by fitting regression curves to the modelled concentrations (natural log of concentrations vs. natural log of distance from source centre), which was then averaged over all models (more details are provided in the supplementary material).for each meteorological dataset (corresponding to each of the EMEP model grid squares). A model-average dispersion kernel (D) for each source height was obtained by taking the mean value of the dispersion model concentration estimates for each kernel grid cell.

These model-average kernels were then multiplied bycombined with the emission data (for each SNAP sector separately in the case of NO$_x$)using a "moving window" approach to obtain the sub-grid concentration estimate (C):

$$C(i,j) = \sum_{s}^{SNAP} \sum_{i'}^{n} \sum_{j'}^{m} E(i',j') \, D'(i-i', j-j')$$

where i and j are the results summed oversub-grid-cell coordinates, s is the entireemission sector, i' and j' are the emission grid cell coordinates, E is the emission rate of the emission grid cell (Mg km$^{-2}$ yr$^{-1}$) and D' an interpolated dispersion kernel (inverse distance squared weighted interpolation of the kernels for the source EMEP grid square and the eight adjacent grid squares). Since the dispersion kernel has a size of $101 \times 101$ grid cells, the values of i' and j' range from i-50 and j-50 to i+50 and j+50, respectively, with the constraint that they lie within the modelling domain (central Scotland or the Netherlands). .

The resulting "sub-grid distributions" provide an estimate of the spatial variability of the concentrations at a $1 \times 1$ km$^2$ resolution, which were then used to "redistribute" the EMEP predictions within each $50 \times 50$ km$^2$ grid square. This step is necessary since the sub-grid modelAQR does not take into account large scale processes such as long-range transport or

chemical transformations of pollutants, processes that are included in the large scale model (the EMEP model, in this case). The simplest way to do this redistribution would be to multiply the sub-grid distributions by the EMEP predictions and then divide by the mean value of the sub-grid distribution for each $50 \times 50$ km$^2$ grid square. This approach conserves the sub-grid distribution for each $50 \times 50$ km$^2$ square and also has the same mean concentration as the EMEP prediction. However, it

5    also could lead to large discontinuities at the edges of the EMEP grid squares if the ratio between the mean of the sub-grid distribution and the EMEP prediction differ greatly from that of adjacent squares. To avoid this problem, the ratio of the EMEP predictions to the mean value of the sub-grid distribution for each $50 \times 50$ km$^2$ square was interpolated on a $1 \times 1$ km$^2$ grid (using a spline interpolation of the values at the centre of each grid square in ArcGIS 10.2 (Environmental Systems Research Institute, Redlands, CA, USA)). The interpolated field was then multiplied by the sub-grid distribution and then the

10    process was repeated over ten iterations. In fact only four-five iterations were necessary to give concentration fields that differed by a maximum of 1%. A more detailed description of the process is provided in the supplementary material.

In order to test the sensitivity of the model to the meteorological data, the above process was repeated with the kernels obtained from the dispersion simulations using the domain-specific meteorological data and with kernels derived from the dispersion simulations using the synthetic meteorological data (more details provided in the supplementary material).

**4 Results**

**4.1 Sub-grid concentration predictions and model evaluation**

Figure 3 shows the sub-grid concentration predictions for NO$_2$ and NH$_3$ for the two domains. (data for the individual domains are provided in Figure S3.1 in the supplementary material). The EMEP concentration fields are also shown for

20    comparison. Table 2 shows the evaluation statistics of the EMEP and sub-gridAQR models for annual mean NO$_2$ concentrations for the Dutch and Scottish monitoring data. In general the sub-grid model performed notably better than, AQR is an improvement on the EMEP model as a result of a consistent underestimation byalone because the latter generally underestimates concentrations (negative NMB). The mean error of the EMEP model is largest for the Scottish dataset with a NMGE of 82% and 70% for the datasets with and without traffic stations, respectively. The model performs worst for the

25    Scottish traffic stations with a mean underestimation of 84%. The EMEP model performs considerably better for the Dutch dataset, with 91% of predictions within a factor of two of the observed values, although this drops to 69% when considering the traffic stations only. The sub-gridAQR model (using $1 \times 1$ km$^2$ emissions) also performed best for the Dutch dataset, with a smaller mean bias and error and a better correlation than the EMEP model alone. However, the EMEP model had a lower mean bias and error for the non-traffic stations. The sub-gridAQR model is also performed better thanan improvement

30    on the EMEP model alone for the Scottish dataset (both with and without traffic stations), as well as for the combined dataset (Netherlands plus Scotland). Similarly to the EMEP model, the sub-grid modelAQR performed worst for the Scottish traffic stations, although notably better thanwas a notable improvement over the EMEP model alone. The use of the lower

resolution emissions actually improved the performance of AQR for some of the statistics (most notably for the non-traffic stations in the Netherlands domain).

Table 3 shows the evaluation statistics of the EMEP and AQR models for annual mean $NH_3$ concentrations for the Dutch and Scottish monitoring data. In general AQR was an improvement on the EMEP model alone, which performed worse for the local monitoring network, as all monitoring locations were within a single EMEP $50 \times 50$ km$^2$ square. The AQR model (using 1 x 1 km$^2$ emissions) also performed worst for this dataset, although its performance was still an improvement on that of the EMEP model alone, as it was for all the datasets except for the National Ammonia Monitoring Network sites in Scotland. The use of the 7 x 7 km$^2$ emissions worsened the performance of AQR for all datasets except for the National Ammonia Monitoring Network sites, for which it had a similar performance to the model using the higher resolution emissions. Figure 4 shows the scatterplots of $NO_2$ and $NH_3$ concentration predictions of the EMEP and AQR models vs. the observed values for all sites in both domains.

**4.2 Sensitivity of sub-grid model predictions to model parameters**

The use of alternative meteorological datasets only had a small effect on the concentration estimates of the AQR model (Fig. 5). The use of domain-specific data from a single location affected the concentration predictions by an average of 6% for  $NO_2$ and 5% for $NH_3$ although differences of up to 23% were found for individual measurement sites. Similarly, the use of the synthetic meteorological data affected concentrations, on average, by 6% and 5% for $NO_2$ and $NH_3$, respectively, with a maximum difference of 28%. Randomising the wind direction data of the domain-specific datasets gave very similar results to those using the synthetic meteorology dataset, with maximum differences of only 1% (not shown). This suggests that the meteorological factor that most influences the  estimates of the AQR model is the wind direction distribution.

The AQR model estimates are also not very sensitive to the $NO_x$ emission height. On average, the effect on the concentration predictions of halving or doubling the emission heights is less than 2%, with a maximum difference of 6% (not shown). This lack of sensitivity to the exact source height  reflects the fact that ground-level sources contribute significantly more to near-source concentrations than elevated sources. Since the concentrations predicted by AQR were not greatly affected by the meteorological data or the emission heights, model performance was very similar (not shown).

**5 Discussion**

**5.1 An improvement, but is it enough?**

These results show that a simple and robust geostatistical approach can be used to improve the EMEP model predictions of $NO_2$ and $NH_3$ annual concentrations. This improvement is not surprising considering the large difference in spatial
5 resolutions (50 km vs. 1 km) and the strong link between short-lived pollutants and the spatial distribution of emissions. In fact, it is worth looking at whether this improvement is mainly a result of the high resolution emissions and has very little to do with the use of short range dispersion estimates. This can be done by repeating the analyses with the $1 \times 1$ km$^2$ grid cell emissions as the initial sub-grid distribution. Figure 6 shows that doing this for $NO_2$ substantially overestimates concentrations for the mid-range of measured values, whereas for $NH_3$ concentrations are
10 substantially underestimated at many sites. The model performance statistics for these simulations show that using just the emissions gives lower values of FAC2 (0.60 vs 0.70 for $NO_2$ and 0.28 vs 0.84 for $NH_3$) and larger bias and error (NMB: 0.36 vs -0.27 for $NO_2$ and -0.36 vs 0.09 for $NH_3$; NMGE: 0.72 vs. 0.41 for $NO_2$ and 0.79 vs 0.27 for $NH_3$). Model error is even larger than that for the EMEP model alone (0.72 vs 0.61 for $NO_2$ and 0.79 vs 0.42 for $NH_3$), which demonstrates that short-range dispersion estimates are necessary for improving on the EMEP model predictions. However, is the improvement of

[revised manuscript text omitted]

The various assumptions and simplifications made in the development of AQR introduce uncertainty in the model estimates. The omission of $NO_x$ photochemistry and the assumption that annual mean $NO_2$ concentrations are linearly correlated with those of $NO_x$ was justified above by the fact that measured concentrations across Europe are approximately linearly correlated ($r^2 = 0.93$). However, a more in-depth analysis of the European measurements shows that if a constant factor is used to estimate $NO_2$ concentrations from the measurements of $NO_x$, the estimated $NO_2$ concentrations differ from the measure values by an average of 16%, which is a small uncertainty compared with the uncertainty in emissions, meteorological conditions, etc. The uncertainty as a result of not modelling the chemical transformation of $NH_3$ (e.g. to particulate ammonium) is more difficult to quantify since the reactions depend on many factors such as the meteorological conditions and the concentrations of other pollutants. However, the fact that the errors (NMGE) in the AQR estimates of $NH_3$ concentrations are of a similar order of magnitude to the errors in the $NO_2$ estimates suggests that the benefits of AQR in handling sub-grid distributions outweigh any chemical impacts. In addition, such errors would be largest far from the

sources, once $NH_3$ concentrations are diluted more to levels comparable to incoming sulphate or $HNO_3$ concentrations. Another source of uncertainty is the omission of deposition processes in the short-range dispersion parameterisations, but wet-deposition has been implicitly included in ACTM predictions, and time-scales for dry deposition are usually far larger than those for sub-grid mixing. Again, given the AQR model has a mean error of 41% and 27% for $NO_2$ and $NH_3$,

5    respectively, the benefits of AQR seem greater than any uncertainty as a result of omitting these processes. Finally, another simplification is the use of a $1 \times 1$ km$^2$ source for parameterising short-range dispersion. In reality sources are generally smaller than this and so this simplification may result in incorrect concentration gradients close to small or linear $NO_x$ sources (e.g. chimney stacks or motorways). However, on average, transport emissions contribute more than 90% of the estimated concentrations, most of which are in urban areas where a $1 \times 1$ km$^2$ source is probably an adequate representation

10   of a dense urban road network. In addition, we rarely know the location of stacks in emission inventories to better than 1 km resolution, and usually with no or very limited information on plume rise and height.

With regards to potential improvements, in addition to the extension to shorter time periods, it also should be possible to incorporate stack

15   parameters (effective emission heights and the contribution of stack emissions to the emissions of a particular grid square) from officially reported data and/or other data sources, if these become more readily available. This would potentially improve concentration estimates close to large stack sources. As shown above, model performance is poorer for sites very close to roads and so the inclusion of a roadside increment model could also improve the model estimates. However, by increasing the complexity of the model, we have to careful not to lose sight of the objective of the AQR model,

20   which is to provide a robust and simple method of post-processing concentrations estimated by a chemical transport model.

The sub-grid approach also has the potential to be applied to other pollutants for which there is a strong relationship between emissions and concentrations. Zhang and Wu (2013) analysed air quality simulations of the CMAQ model to quantify the influence of a range of processes on the atmospheric concentrations of several pollutants. The species that were most strongly influenced by emission processes were: $NH_3$, NO, $NO_2$, $SO_2$, $PM_{2.5}$, $SO_4^{2-}$, elemental carbon, and primary

25   organic aerosol and are, therefore, potential candidates for an extension of the model. The spatial distribution of ozone, a secondary pollutant, cannot be estimated based on emissions but its inverse relationship with $NO_x$ could be exploited to model the sub-grid variability. Apart from concentrations, it may also be possible to develop a sub-grid model for processes such as wet deposition of nitrogen or sulphur, for which high resolution rainfall maps could be used to estimate the sub-grid distributions. Dry deposition of reduced nitrogen could also be modelled using the $NH_3$ concentration distribution and land

30   cover parameters, assuming that most of the deposition is in the form of $NH_3$. Dry deposition of oxidised nitrogen would be more difficult since there is no one dominant species that contributes.

**Conclusions**

The sub-grid spatial variability of the annual mean  NO$_2$ and NH$_3$ concentrations  predicted  by an atmospheric chemistry and transport model can be estimated by combining the predictions with high spatial resolution emission datasets and short-range dispersion fields. This paper describes the development of  the Air Quality
5 Re-gridder (AQR) model and its application to two test domains in Europe. Comparison of  annual mean concentrations estimated by AQR with measured values within both domains shows that the  AQR 
[revised manuscript text omitted]

Simpson, D., Butterbach-Bahl, K., Fagerli, H., Kesik, M., Skiba, U. and Tang, S.: Deposition and emissions of reactive nitrogen over European forests: a modelling study, Atmos. Environ., 40, 5712-5726, 2006.

Simpson, D., Benedictow, A., Berge, H., Bergström, R., Emberson, L. D., Fagerli, H., Flechard, C. R., Hayman, G. D., Gauss, M. and Jonson, J. E.: The EMEP MSC-W chemical transport model–technical description, Atmospheric Chemistry and Physics, 12, 7825-7865, 2012.

Spanton, A.M., Hall, D.J., Dunkerley, F., Griffiths, R.F., Bennett, M.: A Dispersion Model Intercomparison Archive. Proceedings of 9th Int. Conf. on Harmonisation within Atmospheric Dispersion Modelling for Regulatory Purposes, Garmisch-Partenkirchen, Germany, 1–4 June, 2004.

Sutton, M. A., Tang, Y. S., Dragosits, U., Fournier, N., Dore, A. J., Smith, R. I., Weston, K. J. and Fowler, D.: A spatial analysis of atmospheric ammonia and ammonium in the U.K, TheScientificWorldJournal, 1 Suppl 2, 275-286, 2001.

Theobald, M. R., Løfstrøm, P., Walker, J., Andersen, H. V., Pedersen, P., Vallejo, A. and Sutton, M. A.: An intercomparison of models used to simulate the short-range atmospheric dispersion of agricultural ammonia emissions, Environ. Modell. Softw., 37, 90-102, 2012.

Vieno, M., Heal, M. R., Hallsworth, S., Famulari, D., Doherty, R. M., Dore, A. J., Tang, Y. S., Braban, C. F., Leaver, D. , Sutton, M. A., and Reis, S.: The role of long-range transport and domestic emissions in determining atmospheric secondary inorganic particle concentrations across the UK, Atmos. Chem. Phys., 14, 8435-8447, 2014.

Vieno, M., Dore, A. J., Stevenson, D. S., Doherty, R., Heal, M. R., Reis, S., Hallsworth, S., Tarrason, L., Wind, P. , Fowler, D., Simpson, D., and Sutton, M. A.: Modelling surface ozone during the 2003 heat-wave in the UK, Atmos. Chem. Phys., 10, 7963-7978, 2010.

Vogt, E., Dragosits, U., Braban, C. F., Theobald, M. R., Dore, A. J., van Dijk, N., Tang, Y. S., McDonald, C., Murray, S., Rees, R. M. and Sutton, M. A.: Heterogeneity of atmospheric ammonia at the landscape scale and consequences for environmental impact assessment, Environ. Pollut., 179, 120-131, 2013.

von Bobrutzki, K., Braban, C., Famulari, D., Jones, S., Blackall, T., Smith, T., Blom, M., Coe, H., Gallagher, M. and Ghalaieny, M.: Field inter-comparison of eleven atmospheric ammonia measurement techniques, Atmos. Meas. Tech, 3, 91-112, 2010.

Zhang, Y. and Wu, S.: Understanding of the Fate of Atmospheric Pollutants Using a Process Analysis Tool in a 3-D Regional Air Quality Model at a Fine Grid Scale, Atmospheric and Climate Sciences, 3, 18, 2013.

**Table 1: Emission heights used for each main emission sector**

| SNAP Code | Emission sector | Effective emission height used (m) |
|---|---|---|
| 1 | Combustion in energy and transformation industries | 400 |
| 2 | Non-industrial combustion plants | 0 |
| 3 | Combustion in manufacturing industry | 400 |
| 4 | Production processes | 50 |
| 7 | Road transport | 0 |
| 8 | Other mobile sources and machinery | 0 |
| 9 | Waste treatment and disposal | 200 |

**Table 2: Performance evaluation of the EMEP and sub-grid models for annual mean NO$_2$ concentrations. The best performing model for each statistic is highlighted in bold. FAC2 is the fraction of model predictions within a factor of two of the observations, NMB is the normalised mean bias, NMGE is the normalised mean gross error and r is the Pearson correlation coefficient. Shaded cells highlight the model performance for the sub-grid model using the lower resolution emission data.**

| Dataset | n | EMEP model | | | | Sub-grid model | | | | |
|---|---|---|---|---|---|---|---|---|---|---|
| | | FAC2 | NMB | NMGE | r | Emission data | FAC2 | NMB | NMGE | r |
| Netherlands (All) | 43 | 0.91 | -0.24 | 0.31 | 0.54 | 1 × 1 km$^2$ | 0.98 | **0.04** | 0.224 | **0.84 83** |
| | | | | | | 7 × 7 km$^2$ | **1.0** | -0.08 | **0.21** | 0.79 |
| Netherlands (No traffic stations) | 30 | **1.00** | -0.06 | **0.18** | 0.73 | 1 × 1 km$^2$ | 0.97 | 0.07 | 0.229 | **0.86** |
| | | | | | | 7 × 7 km$^2$ | **1.0** | **0.01** | 0.21 | 0.81 |
| Netherlands (Traffic stations only) | 13 | 0.69 | -0.45 | 0.45 | 0.17 | 1 × 1 km$^2$ | **1.0** | **0.00** | 0.17 | **0.58 48** |
| | | | | | | 7 × 7 km$^2$ | **1.0** | -0.18 | 0.21 | 0.32 |
| Scotland (All) | 48 | 0.06 | -0.82 | 0.82 | 0.16 | 1 × 1 km$^2$ | **0.48** | **-0.48** | **0.52** | 0.46 43 |
| | | | | | | 7 × 7 km$^2$ | 0.23 | -0.63 | 0.63 | **0.51** |
| Scotland (No traffic stations) | 11 | 0.27 | -0.70 | 0.70 | 0.40 | 1 × 1 km$^2$ | **0.91** | **-0.07** | **0.30** | 0.80 |
| | | | | | | 7 × 7 km$^2$ | 0.64 | -0.38 | 0.39 | **0.85** |
| Scotland (Traffic stations only) | 37 | 0.00 | -0.84 | 0.84 | 0.05 | 1 × 1 km$^2$ | **0.35** | **-0.54** | **0.55** | 0.50 48 |

|  | | | EMEP | | | | | | | | |
|---|---|---|---|---|---|---|---|---|---|---|
|  | | | | | | | 7 × 7 km² | 0.11 | -0.67 | 0.67 | **0.51** |
| All | 91 | 0.46 | -0.58 | 0.61 | -- | | 1 × 1 km² | **0.0** | **-0.27** | 0.41 | 0.39 |
|  | | | | | | | 7 × 7 km² | 0.59 | -0.40 | 0.46 | 0.27 |

**Table 3: Performance evaluation of the EMEP and sub-grid models for annual mean NH$_3$ concentrations. The best performing model for each statistic is highlighted in bold. FAC2 is the fraction of model predictions within a factor of two of the observations, NMB is the normalised mean bias, NMGE is the normalised mean gross error and r is the Pearson correlation coefficient. Shaded cells highlight the model performance for the sub-grid model using the lower resolution emission data.**

| Dataset | n | EMEP | | | | Sub-grid model | | | | |
|---|---|---|---|---|---|---|---|---|---|---|
|  |  | FAC2 | NMB | NMGE | r | Emission data | FAC2 | NMB | NMGE | r |
| Netherlands | 108 | 0.85 | 0.23 | 0.39 | 0.69 | 1 × 1 km² | **0.92** | **0.10** | 0.24 | **0.84** |
|  |  |  |  |  |  | 7 × 7 km² | 0.90 | 0.28 | 0.40 | 0.71 |
| Scotland – Local network | 21 | 0.52 | -0.47 | 0.65 | -- | 1 × 1 km² | **0.62** | **-0.4** | 0.52 | 0.55 |
|  |  |  |  |  |  | 7 × 7 km² | 0.52 | -0.48 | 0.66 | -- |
| Scotland (National Ammonia Monitoring Network) | 14 | **0.71** | **0.07** | 0.46 | 0.73 | 1 × 1 km² | 0.7 | 0.6 | **0.45** | 0.77 |
|  |  |  |  |  |  | 7 × 7 km² | 0.57 | **0.07** | 0.43 | **0.81** |
| All | 143 | 0.79 | 0.17 | 0.42 | 0.74 | 1 × 1 km² | **0.84** | **0.09** | 0.27 | 0.84 |
|  |  |  |  |  |  | 7 × 7 km² | 0.81 | 0.20 | 0.42 | 0.75 |

5  **Table 4: Number of model acceptability criteria met for each model and evaluation dataset. Shaded cells represent acceptable model performance ( ≥ 3 criteria met).**

| Pollutant | Dataset | | No. of criteria met | | |
|---|---|---|---|---|---|
|  |  |  | EMEP | Sub-Grid (1 × 1 km² emissions) | Sub-Grid (7 × 7 km² emissions) |
| NO$_2$ | Netherlands | All | 5 | 5 | 5 |
|  | Netherlands | No traffic stations | 5 | 5 | 5 |
|  | Netherlands | Traffic stations only | 3 | 5 | 5 |

| | | | | |
|---|---|---|---|---|
| | Scotland | All | 0 | 2 | 1 |
| | Scotland | No traffic stations | 0 | 5 | 3 |
| | Scotland | Traffic stations only | 0 | 2 | 0 |
| | All | | 0 | 3 | 3 |
| $NH_3$ | Netherlands | | 5 | 5 | 5 |
| | Scotland | Local network | 2 | 5 | 2 |
| | Scotland | National Network | 5 | 5 | 5 |
| | All | | 5 | 5 | 5 |

[Figure]

**Figure 1: Spatial distributions of annual emissions of NO$_x$ (left) and NH$_3$ (right), for the Dutch (top) and Scottish (bottom) domains. The EMEP 50 × 50 km$^2$ grid is also shown (in blue).**

[Figure]

**Figure 2: Schematic showing the process of producing the sub-grid concentration predictions from short-range dispersion model simulations and high spatial resolution emission data.**

[Figure]

[Figure]

**Figure 3: Sub-grid model predictions (top row) of annual mean concentrations of NO$_2$ and NH$_3$ for the two domains. EMEP model predictions at a resolution of $50 \times 50$ km$^2$ are shown for comparison (bottom row).**

[Figure]

**Figure 4: Modelled concentrations plotted against measured values for all sites for (a) NO₂ and (b) NH₃. NO₂ traffic stations are indicated by bold symbol outlines. Plot data provided in the supplementary material.**

[Figure]

**Figure 5: Modelled concentrations plotted against measured values for all sites for (a) NO$_2$ and (b) NH$_3$ using  the original meteorology (as in Figure 4) and using the domain-specific and synthetic meteorological datasets.**

[Figure]

**Figure 6: Modelled concentrations plotted against measured values for all sites for (a) NO$_2$ and (b) NH$_3$ using the original sub-grid parameterisation (Emission plus dispersion) and using just the spatial distribution of emissions as the sub-grid distribution (Emission only).**